# Rescue of imprinted genes by epigenome editing in human cellular models of Prader-Willi syndrome

Akisa Nemoto[1,2,3], Kent Imaizumi [1,2,6,7], Fuyuki Miya [4], Yuka Hiroi [3], Mamiko Yamada[4], Hirosato Ideno[1], Shinji Saitoh [5], Kenjiro Kosaki[4], Hironobu Okuno [1,2,3,4] ✉ & Hideyuki Okano [1,2] ✉

Prader-Willi syndrome (PWS) is a genomic imprinting disorder caused by the loss of function of the paternal chromosome 15q11-13, resulting in a spectrum of symptoms associated with hypothalamic dysfunction. PWS patients lack the expression of paternally expressed genes (PEGs) in the 15q11-13 locus but possess an epigenetically silenced set of these genes in the maternal allele. Thus, activation of these silenced genes can serve as a therapeutic target for PWS. Here, we leverage CRISPR-based epigenome editing system to modulate the DNA methylation status of the PWS imprinting control region (PWS-ICR) in induced pluripotent stem cells (iPSCs) derived from PWS patients. Successful demethylation in the PWS-ICR restores the PEG expression from the maternal allele and reorganizes the methylation patterns in other PWS-associated imprinted regions beyond the PWS-ICR. Remarkably, these corrected epigenomic patterns and PEG expression are maintained following the differentiation of these cells into hypothalamic organoids. Finally, the single-cell transcriptomic analysis of epigenome-edited organoids demonstrates a partial restoration of the transcriptomic dysregulation observed in PWS. This study highlights the utility of epigenome editing technology as a therapeutic approach in addressing PWS and potentially other imprinting disorders.

Prader-Willi syndrome (PWS) results from the loss of function of paternally expressed genes (PEGs) within the chromosome 15q11-13 imprinted region[1]. Primary clinical manifestations of PWS include hyperphagia, hypogonadism, and short stature due to growth hormone deficiency, and these phenotypes have been thought to be closely related to hypothalamic dysfunction; however, little is known about the cellular and molecular pathophysiology. The 15q11-13 imprinted region has several PEGs, including *MKRN3*, *MAGEL2*, *NDN*, *SNRPN*, *SNORD116*, *IPW*, and *SNORD115*. Among these, *SNORD116* plays

a critical role in PWS pathology, as indicated by patients with microdeletions affecting this specific locus[2–4], while the involvement of the other genes remains debated.

PWS patients harbor PWS-associated genes on the maternal allele, but they are silenced. This suggests potential therapies aimed at restoring the expression of these genes from the maternal allele. The maternal allele-specific silencing is tightly regulated by the epigenetic status of the PWS imprinting control region (PWS-ICR), located within the promoter and exon 1 regions of *SNRPN*[5,6], hence, epigenetic

[1]Department of Physiology, Keio University School of Medicine, Tokyo, Japan. [2]Keio University Regenerative Medicine Research Center, Kawasaki, Japan. [3]Department of Pediatrics and Adolescent Medicine, Tokyo Medical University, Tokyo, Japan. [4]Center for Medical Genetics, Keio University School of Medicine, Tokyo, Japan. [5]Department of Pediatrics and Neonatology, Nagoya City University Graduate School of Medical Sciences, Nagoya, Japan. [6]Present address: Department of Psychiatry and Behavioral Sciences, Stanford University, Stanford, CA, USA. [7]Present address: Stanford Brain Organogenesis, Wu Tsai Neurosciences Institute, Stanford University, Stanford, CA, USA. ✉e-mail: okuno.hironobu.6v@tokyo-med.ac.jp; hidokano@keio.jp

manipulation targeting the PWS-ICR can serve as a potential method to reactivate the silenced maternal expression of PWS-associated genes. Studies on epigenetic manipulation have been reported by small molecules, such as histone methyltransferase inhibitors[7]. However, these epigenetic manipulation drugs may exert genome-wide effects, presenting challenges for precise locus-specific targeting. Furthermore, these interventions did not fully activate the expression of PEGs up to the level of healthy individuals. On the other hand, recent advancements in CRISPR-based epigenome editing present promising opportunities[8]. These technologies manipulate the epigenetic status of specific genomic regions by introducing guide RNAs (gRNAs). The CRISPR/dCas9-Suntag-TET1 system is one of those epigenome editing technologies and utilizes a mutated form of Cas9 (dead/deactivated Cas9) fused with Suntag, recruiting multiple copies of DNA demethylase TET1[9]. This approach facilitates efficient DNA demethylation in the targeted region while minimizing off-target epigenetic alterations. Thus, the application of CRISPR-based epigenome editing holds potential for reactivating silenced PWS-associated genes on the maternal allele, offering a promising avenue for therapeutic exploration.

In this study, we utilized the CRISPR/dCas9-Suntag-TET1 system to target the PWS-ICR in induced pluripotent stem cells (iPSCs) derived from PWS patients. The CpG islands in the maternal PWS-ICR were successfully demethylated, and the expression of PWS-associated PEGs was restored from the maternal allele. The epigenetic manipulation and the activated maternal PEG expression were maintained after differentiation into hypothalamic organoids. Furthermore, these epigenome-edited organoids demonstrated the amelioration of some transcriptomic signatures linked to PWS pathology. By using an iPSC-based human cellular model of PWS, this study provides proof of principle for CRISPR-mediated epigenome editing to treat PWS.

## Results

### Restoration of *SNRPN* gene expression through CRISPR-mediated epigenome editing

We aimed to modulate the DNA methylation status of the PWS-ICR in iPSCs using the CRISPR/dCas9-Suntag-TET1 system[9] (Fig. 1a). We used iPSCs derived from four different patients: deletion type, 2PWS8[10] and 5PWS10; maternal uniparental disomy (mUPD), 3PWS14; imprinting defect, 4PWS1 (Supplementary Fig. 1a–d). Five guide RNAs (gRNAs) targeting the PWS-ICR (Supplementary Fig. 1e, f) were delivered to these iPSCs by lentivirus, and plasmids encoding dCas9-Suntag-TET1 components (dCas9-GCN4s and scFv-sfGFP-TET1CD) were transfected by lipofection (Supplementary Fig. 1g, h). The successful transient overexpression of dCas9-Suntag-TET1 components was confirmed by sfGFP fluorescence and RT-qPCR (Supplementary Fig. 2a, b). As the PWS-ICR encompasses the promoter and exon 1 regions of *SNRPN*, one of PEGs in 15q11-13, and its methylation status regulates *SNRPN* transcription[5], we used *SNRPN* expression as an indicator of epigenome editing (Fig. 1a). Following the transfection, *SNRPN* expression gradually increased in a week (Supplementary Fig. 2c). We then expanded single-cell-derived clones from transfected cells and found some clones with high *SNRPN* expression (Supplementary Fig. 2d). Sanger sequence and short tandem repeat analyses showed that the *SNRPN* upregulation in these clones did not result from genomic mutations of targeted regions nor from contamination with healthy control iPSCs (Supplementary Fig. 2e, f). Remarkably, the *SNRPN* expression level of these clones was comparable to that of healthy control lines (Fig. 1b, Supplementary Fig. 3a), and was sustained after repeated passages (Fig. 1c). Importantly, the transfection process did not impact the expression of pluripotency markers (Supplementary Fig. 3b). These results suggest that *SNRPN* expression was rescued through epigenome editing targeting the PWS-ICR.

### Demethylation of PWS-ICR and off-target effects by epigenome editing

The DNA methylation status of the PWS-ICR was examined by genomic qPCR following methylation-sensitive restriction enzyme digestion (Supplementary Fig. 3c–e), targeted bisulfite genomic sequencing, and nanopore long-read sequencing, respectively. As expected, control iPSCs exhibited around 50% methylation (Fig. 1d–f, Supplementary Fig. 3f–i), consistent with the fact that this region is maternally methylated and paternally unmethylated[1]. On the other hand, all iPSC lines from PWS patients displayed 100% methylation, whereas epigenome-edited clones showed demethylation down to 0% in deletion type iPSCs (2PWS8 and 5PWS10) (Fig. 1d–f, Supplementary Fig. 3f–h). In addition, iPSC lines with mUPD (3PWS14) and imprinting defect (4PWS1), both carrying two copies of the silenced 15q11-13 region, demonstrated demethylation to 50% (Supplementary Fig. 3f). As for mUPD (3PWS14), we conducted additional analysis on two clones that had around two times higher expression of *SNRPN* than the control (Supplementary Fig. 2d). These two clones were fully demethylated to 0% (Supplementary Fig. 3g), suggesting that some clones exhibited monoallelic demethylation, while others had biallelic demethylation. These findings indicate the successful demethylation of the PWS-ICR by epigenome editing.

Evaluating the potential off-target effects is crucial in the assessment of epigenome editing. To explore these effects, we identified potential off-target loci sharing sequence homology with gRNA targets by the in silico tool CRISPRdirect[11] (Supplementary Fig. 4a). Through whole-genome nanopore sequence and genomic qPCR analysis following methylation-sensitive enzyme digestion, we observed that these identified candidate sites did not exhibit methylation alterations induced by epigenome editing (Supplementary Fig. 4b–d). Additionally, we performed bulk RNA sequencing analysis and found that epigenome editing did not affect the expression of genes located nearest these off-target candidate sites (Supplementary Fig. 4e). These data suggest that our CRISPR-mediated epigenome editing has no significant off-target effects, as demonstrated in previous reports[9,12,13].

### Rescued expression and epigenomic reorganization of other PWS-associated imprinted regions by epigenome editing of PWS-ICR

We next investigated the impact of PWS-ICR-targeted epigenome editing on the expression of other PWS-associated genes (Fig. 1g, Supplementary Fig. 5a, b). The expression of *SNORD116* and *IPW*, which are processed from the intron of the *SNHG14* gene and part of *SNHG14* transcript, respectively, was upregulated and comparable to the healthy control iPSCs (Fig. 1h, Supplementary Fig. 6a). The expression of *MAGEL2* was also upregulated in epigenome-edited iPSCs (Fig. 1i, Supplementary Fig. 6b), and nanopore long-read sequencing confirmed that the promoter region of *MAGEL2* was demethylated by epigenome editing (Fig. 1j, Supplementary Fig. 6c, f). These results indicate that epigenome editing targeting the PWS-ICR unsilenced the expression of not only *SNRPN* but also other PWS-associated genes.

Contrary to *MAGEL2*, the gene expression of another imprinted gene, *NDN*, remained unaltered following epigenome editing (Fig. 1k, Supplementary Fig. 6d). Correspondingly, the promoter region of *NDN* did not display any noticeable demethylation (Fig. 1l, Supplementary Fig. 6e, f). These findings suggest that, unlike certain genes whose expression was activated, *NDN* remained unaffected by PWS-ICR-targeted epigenome editing, indicating a partial or lack of activation for some imprinted genes via this editing approach.

### Dynamics of imprinted gene reorganization

CRISPR/dCas9-Suntag-TET1 system induces demethylation within about 200 bp of the target sites[9], and *MAGEL2* promoter region is located more than 1 Mb away from the PWS-ICR, suggesting that the observed demethylation in *MAGEL2* was not a direct consequence of

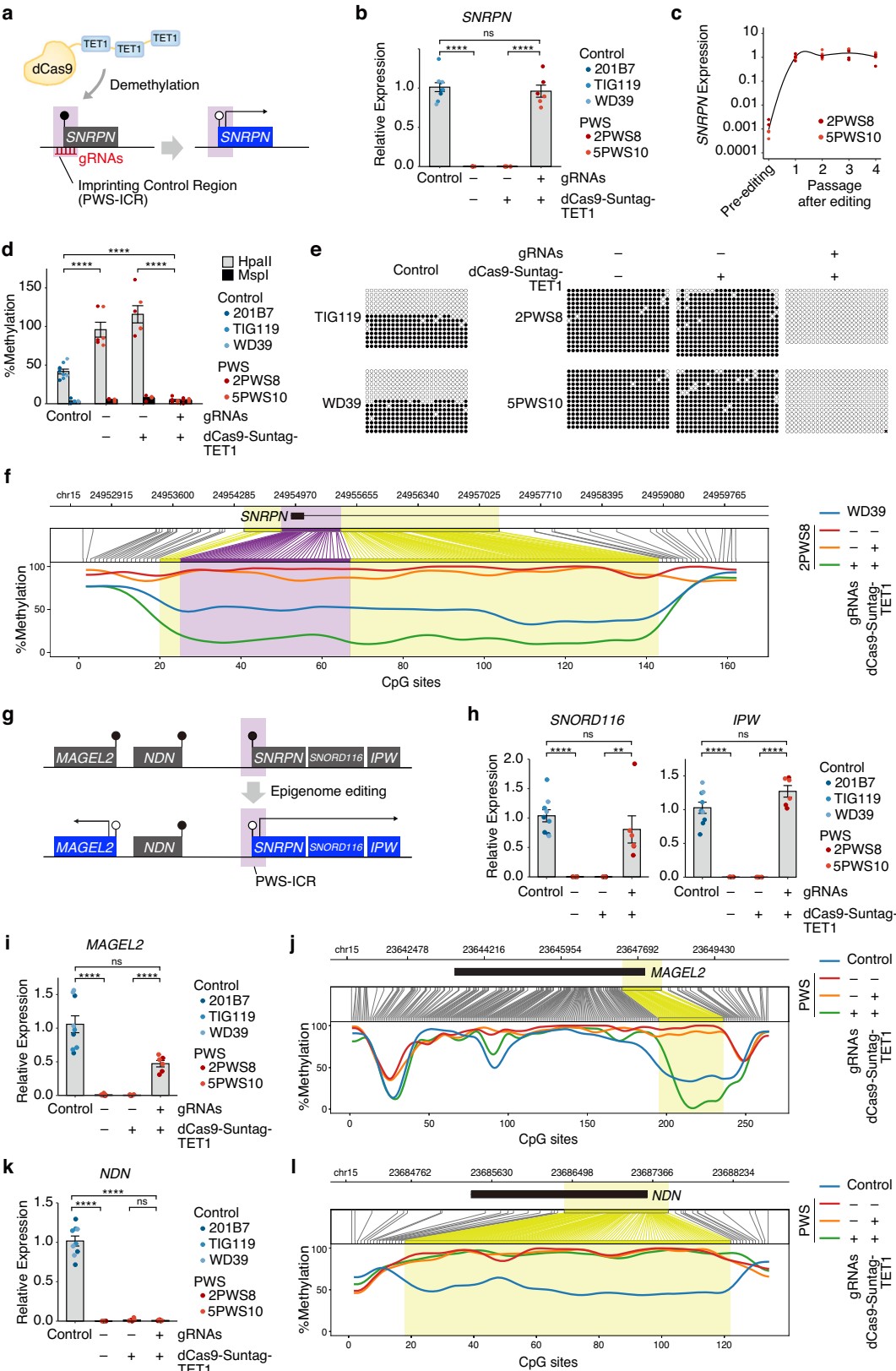

CRISPR-based epigenome editing specifically targeting the PWS-ICR. To further address the mechanism of *MAGEL2* reactivation, we analyzed the methylation dynamics during the epigenome editing process. We performed gRNA lentiviral infection and dCas9-Suntag-TET1 lipofection in PWS iPSCs and then maintained the cells in bulk without single-cell cloning, allowing us to observe the transition dynamics right

after transfection (Supplementary Fig. 7a). We observed that PWS-ICR demethylation occurred immediately after dCas9-Suntag-TET1 transfection. In contrast, *MAGEL2* demethylation was observed subsequently, particularly at least a few passages after the transfection (Supplementary Fig. 7b), by which time dCas9-Suntag-TET1 had already been removed (Supplementary Fig. 2b). We also detected a

**Fig. 1 | Epigenome editing of PWS patient-derived iPSCs targeting 15q11-13 imprinting control region. a** Schematic illustrating the strategy of CRISPR-mediated epigenome editing to target PWS-ICR. Demethylation of PWS-ICR induces the transcription of *SNRPN*. **b** RT-qPCR analysis of *SNRPN* expression (Mean ± SEM; n = 6–9 experiments, 2–6 iPSC line/clones; One-way ANOVA with Tukey correction; $F_{3, 23}$ = 509.3; ****P < 0.0001). **c** Temporal expression changes of *SNRPN* expression in epigenome-edited PWS iPSCs (n = 6 clones from 2 iPSC lines). **d** Methylation status of PWS-ICR in control iPSCs, PWS iPSCs, and epigenome-edited PWS iPSCs, measured by genomic qPCR analysis following methylation-sensitive enzyme digestion (Mean ± SEM; n = 6–9 experiments, 2–6 iPSC line/clones; One-way ANOVA with Tukey correction; $F_{7, 46}$ = 82.03, ****P < 0.0001). See also Supplementary Fig. 3c–f. **e** Bisulfite genomic sequencing of PWS-ICR. White and black circles indicate unmethylated and methylated CpGs, respectively. **f** Methylation status of PWS-ICR by long-read sequencing analysis. The purple highlighted region indicates the gRNA-targeted sites, and the yellow indicates differentially methylated regions. **g** Schematic illustrating the methylation and expression status of PWS-associated imprinted genes. All genes were methylated and silenced in PWS iPSCs, and PWS-ICR-targeted epigenome editing activated the expression of *SNPRN*, *SNORD116*, *IPW*, and *MAGEL2*, but not *NDN*. The methylation status of *MAGEL2*, but not *NDN*, was reorganized. RT-qPCR analysis of the expression of *SNORD116*, *IPW* (**h**) and *MAGEL2* (**i**) (Mean ± SEM; n = 6–9 experiments, 2–6 iPSC line/clones; One-way ANOVA with Tukey correction; *SNORD116*, $F_{3, 23}$ = 139.3; *IPW*, $F_{3, 23}$ = 266.5; *MAGEL2*, $F_{3, 23}$ = 39.6; **P = 0.0049, ****P < 0.0001). **j** Methylation status of *MAGEL2* region by long-read sequencing analysis. The yellow highlighted region indicates differentially methylated regions in *MAGEL2* promoter. **k** RT-qPCR analysis of the expression of *NDN* (Mean ± SEM; n = 6–9 experiments, 2–6 iPSC line/clones; One-way ANOVA with Tukey correction; $F_{3, 23}$ = 35.06; ****P < 0.0001). **l** Methylation status of *NDN* region by long-read sequencing analysis. The yellow highlighted region indicates differentially methylated regions in *NDN* promoter. ns not significant.

time lag of the expression upregulation between *SNRPN* and *MAGEL2* (Supplementary Fig. 7c). These delays in demethylation and gene expression of *MAGEL2* suggest that the demethylation of *MAGEL2* is not a consequence of off-target effects from dCas9-Suntag-TET1. Rather, these results indicate that the PWS-ICR demethylation via epigenome editing triggered secondary demethylation in other PWS-associated imprinted regions, leading to the restoration of the expression of neighboring PEGs.

## PEG expression and DNA methylation status during the hypothalamic organoid differentiation

As most of the clinical symptoms of PWS are associated with the hypothalamus, we directed the differentiation of epigenome-edited 2PWS8 iPSCs into hypothalamic organoids (Fig. 2a, b). The epigenome editing did not affect organoid generation (Fig. 2c), and immunostaining analyses showed the hypothalamus identity of differentiating organoids (Fig. 2d, e). We confirmed that the activation levels of PWS-associated PEGs in epigenome-edited organoids remained comparable to those in healthy control iPSC-derived organoids (Fig. 2f). Moreover, nanopore long-read sequencing revealed the sustained demethylation of the PWS-ICR and its neighboring imprinted regions following organoid differentiation (Fig. 2g, Supplementary Fig. 8a, b). These observations indicate that the activation of PEG expression and the demethylation of the imprinted regions, initially observed in undifferentiated iPSCs, persist consistently following the differentiation into hypothalamic organoids.

## Activation of neuron-specific imprinted gene expression in epigenome-edited hypothalamic organoids

At the iPSC stage, *SNORD115* expression remained undetectable, even in control iPSCs, consistent with a previous report that *SNORD115* functions as a neuron-specific snoRNA[14] (Fig. 3a). Upon differentiation, *SNORD115* expression was drastically upregulated in control iPSC-derived hypothalamic organoids, but not in PWS organoids (Fig. 3b). Epigenome-edited PWS organoids exhibited markedly elevated expression levels comparable to those in control organoids, suggesting that the epigenome editing reactivated neuron-specific PEGs, as well as pan-tissue expressed PEGs.

In addition to *SNORD115*, *UBE3A-ATS* is known as neuron-specific PEG in the 15q11-13 region. *UBE3A-ATS* encodes an lncRNA antisense to *UBE3A*, and the expression of *UBE3A* from the paternal allele is silenced as a result of transcriptional interference in cells expressing *UBE3A-ATS*. This phenomenon results in the neuron-specific maternal expression of *UBE3A*[15,16]. Epigenome editing triggered the activation of *UBE3A-ATS* expression in PWS organoids (Fig. 3c). Subsequently, we observed a concurrent downregulation of *UBE3A* in edited organoids (Fig. 3c).

We also investigated another PWS-associated PEG, *MKRN3*. Similar to *SNORD115* and *UBE3A-ATS*, *MKRN3* expression was not detected at the iPSC stage, and the expression was upregulated following the

organoid differentiation from control iPSCs (Fig. 3d). While the imprinted DNA methylation pattern in *MKRN3* promoter region was not observed in both control and PWS iPSCs, control iPSCs acquired a 50% methylation pattern after organoid differentiation (Fig. 3e, f, Supplementary Fig. 9a, b). In contrast, PWS iPSCs did not upregulate *MKRN3* expression or change DNA methylation pattern after differentiation. Remarkably, epigenome editing partially restored the expression in organoids (Fig. 3d), and genomic qPCR following restriction enzyme digestion revealed that DNA methylation levels were decreased to 80% (Fig. 3f). However, nanopore sequencing data did not show a significant difference, likely due to the relatively low coverage (Fig. 3e). These findings imply that *MKRN3* imprinting is at least partially unsilenced by PWS-ICR-targeted epigenome editing.

## Characterization of gene expression changes in epigenome-edited organoids

To comprehensively characterize organoids with epigenome editing, we performed single-cell RNA sequencing (scRNA-seq) at days 63–66 of differentiation. Cells were classified into cycling cells, progenitors, and neurons, and the cell diversity was overall similar between gRNAs(+) and (−) organoids (Fig. 4a–d). Consistent with our immunostaining analysis (Fig. 3d, e), we confirmed the hypothalamic identity of our organoids by comparing our scRNA-seq data with a previous dataset of iPSC-derived hypothalamic organoid[17] (Supplementary Fig. 10a, b), and by analyzing the similarity between these organoids and mouse embryonic brains using VoxHunt[18] (Fig. 4e and Supplementary Fig. 10c). We next compared the gene expression in each cluster between gRNAs(+) and (−) organoids. Using gene set enrichment analysis (GSEA), we found enrichment for specific biological processes in each cell type (Fig. 4f). In the neuron cluster, genes upregulated in gRNAs(+) organoids showed enrichment for synapse functions. In all the clusters, gRNAs(+)-downregulated genes were associated with ribosome biogenesis and function. Given that the downregulation of synapse-related genes and the upregulation of ribosomal genes were observed in postmortem hypothalamus samples of PWS patients compared with control samples[19], these observations imply that the epigenome editing ameliorates some gene expression signatures of PWS pathology.

To further demonstrate the relationship between the gene expression changes induced by epigenome editing and the PWS pathological characteristics, we employed the improved rank–rank hypergeometric overlap (RRHO[20]) test. This analysis aimed to assess the similarity between the alterations in our edited organoids and the changes observed in PWS versus control postmortem hypothalamus samples and iPSC-derived hypothalamic organoids (Fig. 4g–i). Notably, there is a concordant overlap of gRNAs(+)-upregulated genes and PWS-downregulated genes. In addition, a subset of common differentially expressed genes exhibited comparable fold changes (Fig. 4j). These observations suggest that epigenome editing at least partially reverses the gene expression dysregulation observed in PWS.

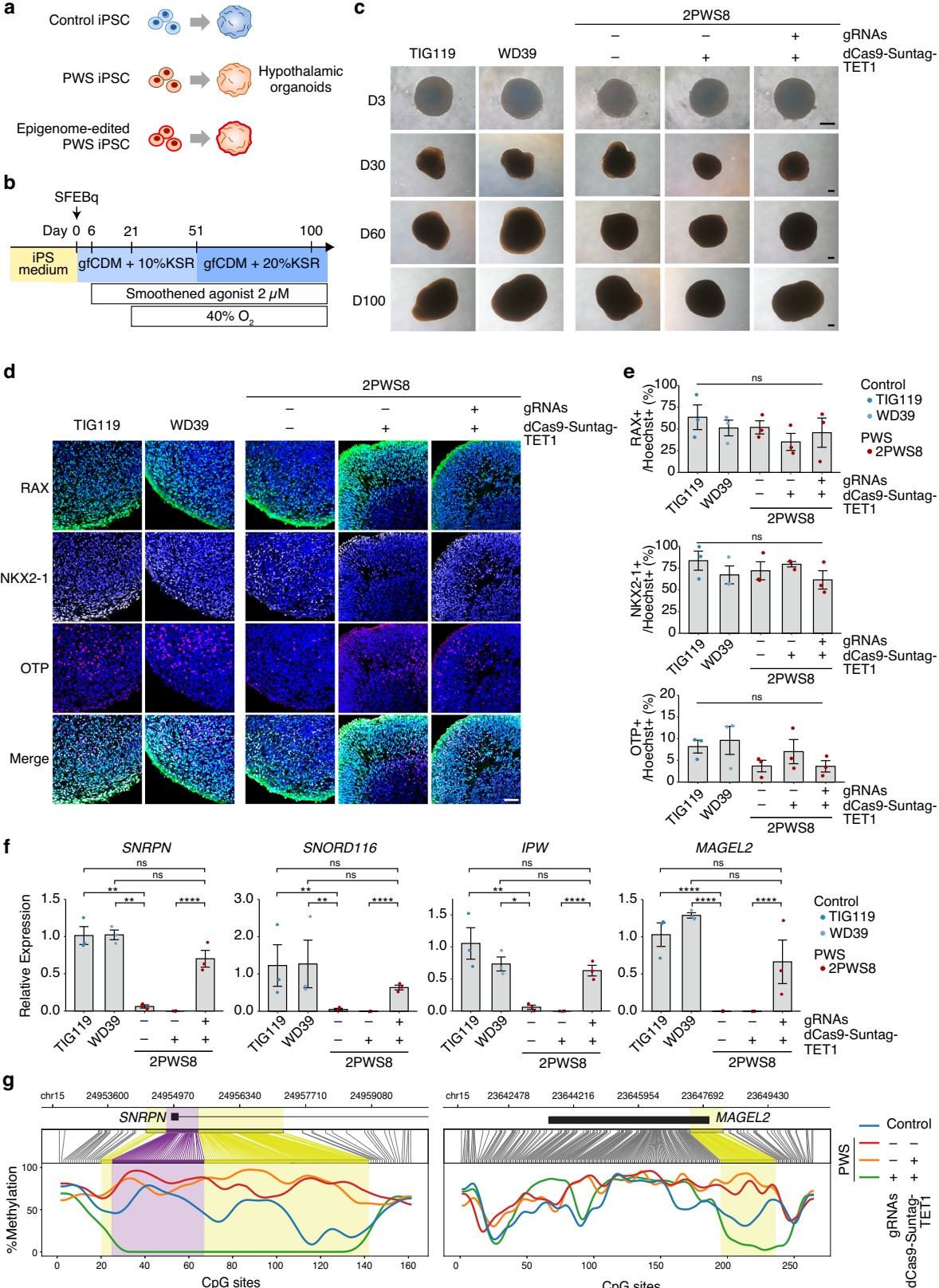

## Neuronal activity in epigenome-edited organoids

In addition to transcriptomic characteristics, a previous paper suggests an altered neuronal activity in PWS cells[17]. To investigate whether this functional consequence can be reversed by epigenome editing, we performed calcium imaging of hypothalamic organoids from control, PWS, and epigenome-edited iPSCs (Supplementary Fig. 11a, b). We observed decreased spontaneous activity in PWS organoids compared with control organoids (Supplementary Fig. 11c). Notably, epigenome-edited organoids exhibited increased activity that was comparable to that of control organoids, suggesting the epigenome editing strategy is effective for functional restoration.

**Fig. 2 | Maintenance of epigenome editing effect in hypothalamic differentiation. a** Schematic illustrating the differentiation of control iPSCs, PWS iPSCs, and epigenome-edited PWS iPSCs into hypothalamic organoids. **b** Protocol to generate hypothalamic organoids from iPSCs. **c** Bright-field images of hypothalamic organoids from control iPSCs, PWS iPSCs, and epigenome-edited PWS iPSCs. Scale bar, 200 μm. We repeated the differentiation at least three times from each iPSC line/clones with similar results. **d** Immunohistochemistry images showing the expression of hypothalamus markers in organoids at day 30. Scale bar, 50 μm. **e** Percentage of cells positive for hypothalamus markers in organoids at day 30 (Mean ± SEM; n = 3 differentiations, 1−3 iPSC line/clones; One-way ANOVA; RAX,

$F_{4, 10} = 0.7394$; NKX2-1, $F_{4, 10} = 0.8817$ for NKX2-1; OTP, $F_{4, 10} = 1.503$). **f** RT-qPCR analysis of the expression of imprinted genes in organoids at day 60 (Mean ± SEM; n = 3 differentiations, 1−3 iPSC line/clones; One-way ANOVA with Tukey correction; *SNRPN*, $F_{4, 10} = 79.78$, **P = 0.0012, ****P < 0.0001; *SNORD116*, $F_{4, 10} = 60.83$, **P = 0.0013, ****P < 0.0001; *IPW*, $F_{4, 10} = 39.50$, *P = 0.0101, **P = 0.0052, ****P < 0.0001; *MAGEL2*, $F_{4, 10} = 154.0$, ****P < 0.0001). **g** Methylation status of PWS-ICR and *MAGEL2* region in organoids at day 100 by long-read sequencing analysis. The purple highlighted region indicates the gRNA-targeted sites, and the yellow indicates differentially methylated regions. ns not significant.

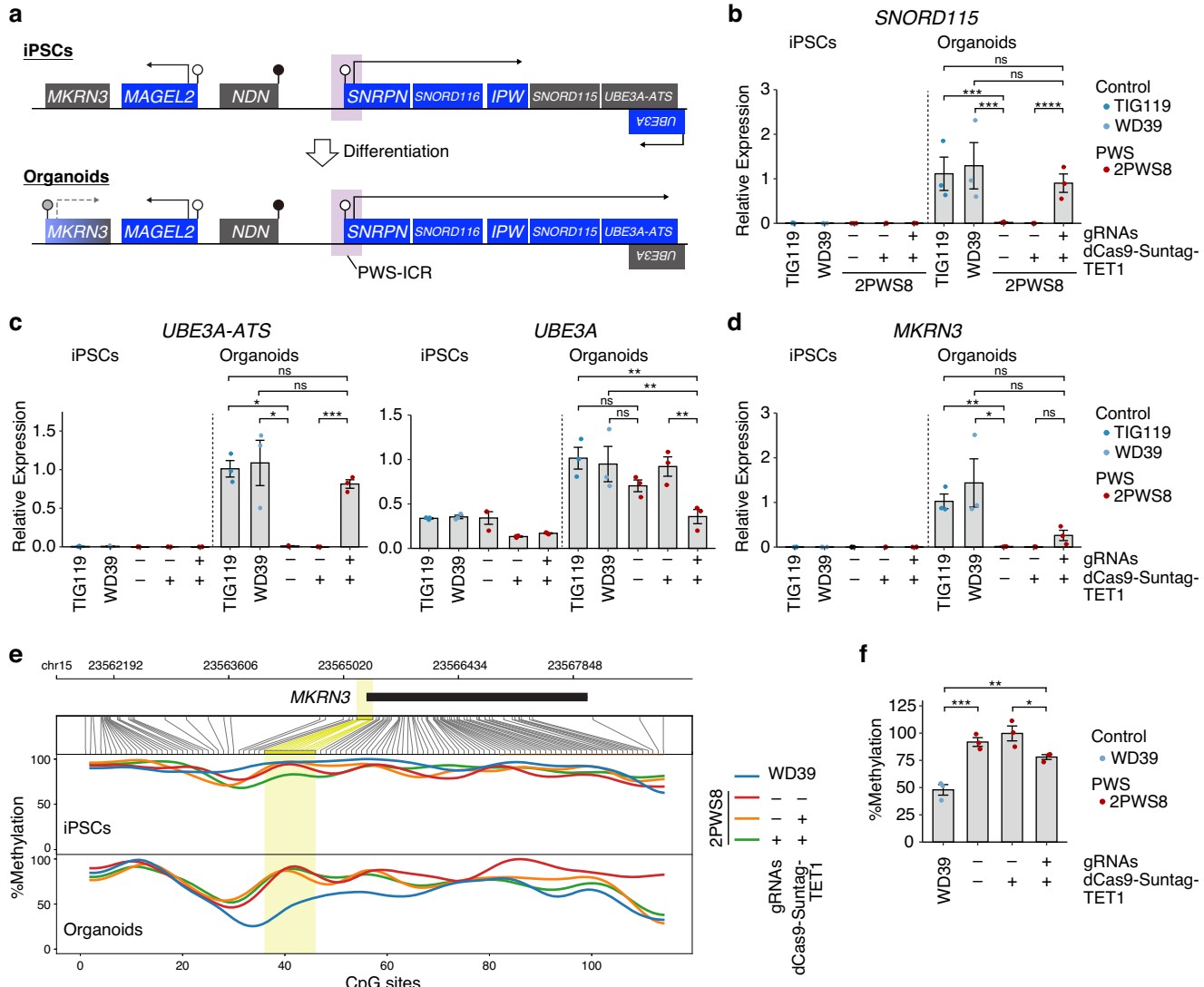

**Fig. 3 | Expression of neuron-specific imprinted genes after hypothalamic differentiation. a** Schematic illustrating the methylation and expression status of PWS-associated imprinted genes during neural differentiation from epigenome-edited iPSCs. At the iPSC stage, epigenome editing activated the expression of *SNORD116*, *IPW*, and *MAGEL2*. Upon neural differentiation, the downstream genes, *SNORD115* and *UBE3A-ATS* were activated. *MKRN3* expression was partially activated. RT-qPCR analysis of the expression of *SNORD115* (**b**), *UBE3A-ATS* and *UBE3A* (**c**), and *MKRN3* (**d**) in iPSCs and organoids at day 60 (Mean ± SEM; iPSCs, n = 3 experiments, 1−3 iPSC line/clones; Organoids, n = 3 differentiations, 1−3 iPSC line/ clones; One-way ANOVA with Tukey correction; *SNORD115*, $F_{9, 20} = 43.62$, ***P = 0.0002, ****P < 0.0001; *UBE3A-ATS*, $F_{9, 20} = 27.85$, *P = 0.0229 [TIG119 vs −/−],

0.0235 [WD39 vs −/−], ***P = 0.0001; *UBE3A*, $F_{9, 20} = 23.67$, **P = 0.0012 [TIG119 vs +/+], 0.0032 [WD39 vs +/+], 0.0033 [−/+ vs +/+]; Brown-Forsythe and Welch ANOVA with Dunnett's T3 correction; *MKRN3*, *P = 0.0144, **P = 0.0011). **e** Methylation status of *MKRN3* region in iPSCs (upper) and organoids at day 100 (bottom) by long-read sequencing analysis. The yellow highlighted region indicates differentially methylated regions in *MKRN3* promoter. **f** Methylation status of *MKRN3* region in organoids at d60, measured by genomic qPCR analysis following methylation-sensitive enzyme digestion (Mean ± SEM; n = 3 differentiations, 1−3 iPSC line/ clones; One-way ANOVA with Tukey correction, $F_{3, 8} = 22.79$, ***P = 0.0009, **P = 0.0091, *P = 0.0495). ns not significant.

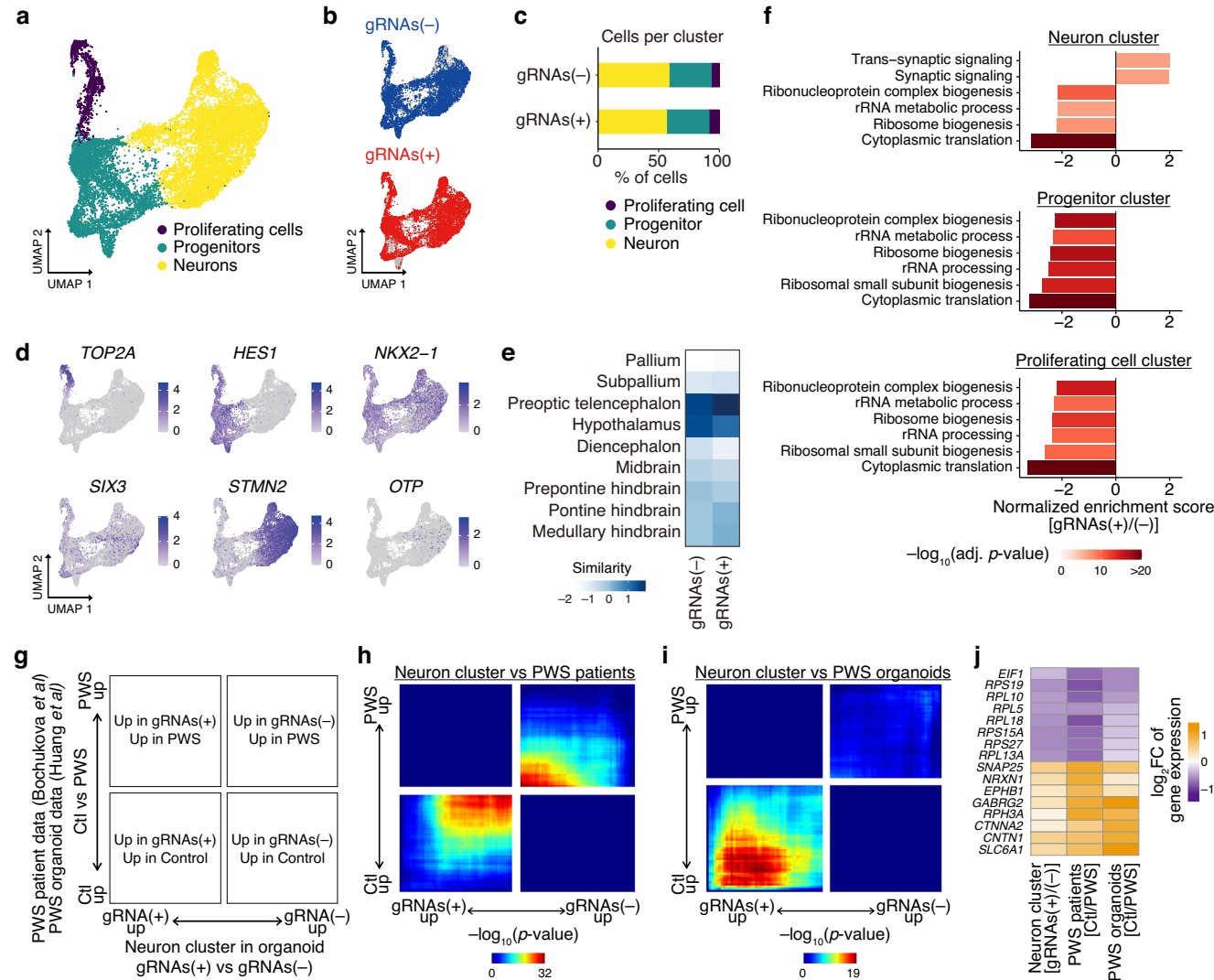

**Fig. 4 | Transcriptomic analysis of epigenome-edited organoids. a** UMAP plot of hypothalamic organoids colored by cell type clusters. **b** UMAP plots of hypothalamic organoids colored by the epigenome editing status. **c** Cell type proportions colored by cell type clusters. **d** UMAP plots showing marker gene expression. **e** VoxHunt spatial brain mapping onto E15.5 mouse brains based on the ISH data from the Allen Developing Mouse Brain Atlas. **f** Top 6 GO terms enriched in genes ranked by fold change between gRNA(+) and (−) samples using gene set enrichment analysis (GSEA). **g** Schematic of the RRHO2 plot. Genes are ordered by fold change between gRNA(+) vs (−) in organoid samples and control vs PWS in PWS patient/organoid datasets, with the most upregulated gene in the lower left corner. RRHO2 plots comparing the gene expression change of epigenome-edited organoids to that of postmortem PWS hypothalamus (**h**) and that of hypothalamic organoids derived from PWS iPSCs (**i**). The color indicates the p-value from two-sided hypergeometric tests with Benjamini–Hochberg correction, which measure the significance of overlap between ranked gene lists. **j** Heatmap showing fold changes in gene expression in epigenome-edited organoids, postmortem PWS hypothalamus, and hypothalamic organoids derived from PWS iPSCs.

## Epigenome editing in PWS organoids

We sought to directly reactivate the PWS-ICR in differentiating organoids by CRISPR-mediated epigenome editing, potentially allowing for more effective therapeutic interventions for PWS. gRNAs-mCherry and dCas9-Suntag-TET1 components (dCas9-GCN4s and scFv-sfGFP-TET1CD) were delivered to organoids derived from PWS iPSCs by lentivirus and lipofection, respectively (Supplementary Fig. 12a, b). We observed a ~6-fold increase in *SNRPN* expression (Supplementary Fig. 12c), though it did not reach the levels observed in healthy control lines. These results suggest that epigenome editing in differentiating cells can partially reactivate *SNPRN*.

## Discussion

We applied CRISPR-based epigenome editing technology to human cellular models of PWS. DNA demethylation was introduced at the PWS-ICR in patient-derived iPSCs, resulting in the reactivation of PWS-associated PEGs from the maternal allele. This reactivation persisted during the differentiation into hypothalamic organoids, leading to a reversal of gene expression patterns associated with PWS pathology. Our study provides proof of concept for CRISPR-mediated epigenome editing to treat PWS, and also suggests the potential applicability of this strategy in addressing other imprinting diseases.

Activation of PEGs on the maternal allele has been explored in several previous studies. Histone methyltransferase EHMT2 inhibitors, UNC0638 and UNC0642, showed efficacy in restructuring histone modifications within the 15q11-13 region, leading to the activation of PEG expression from maternal allele[7]. Similar effects were also observed when histone methyltransferase *SETDB1* was knocked down[21]. Although these approaches exhibit therapeutic promise for treating PWS, EHMT2 and SETDB1 are involved not only in the 15q11-13 locus but also in other genomic regions, potentially leading to off-target effects. In contrast, CRISPR-mediated editing technology offers locus-specific manipulation. Our study, along with recent studies, demonstrated the efficacy and accuracy of CRISPR-mediated

epigenome editing[12,13,22]. The precision of CRISPR-mediated epigenome editing would be beneficial for the clinical therapeutics of PWS patients.

Our epigenome editing showed that demethylation at the PWS-ICR induced the delayed secondary demethylation in adjacent imprinted regions, such as the *MAGEL2* promoter. This observation suggests that the methylation status of the PWS-ICR governs the locus-specific methylation of neighboring imprinted regions. This hierarchical regulation has been expected from the observation that patients with microdeletions at the PWS-ICR had full methylation status in neighboring imprinted regions[23].

The scope of CRISPR-mediated epigenome editing extends beyond PWS, holding promising implications for addressing other imprinting diseases. This approach has also been demonstrated in other epigenetic diseases, including Fragile X syndrome and Rett syndrome[12,13]. Additionally, recent investigations revealed the role of aberrant DNA methylation in cancers and age-related disorders[24,25]. Thus, the findings presented in this study not only advance the understanding and potential treatment avenues for PWS but also contribute to epigenome editing therapies for a wide spectrum of diseases.

## Limitations of study
Our CRISPR-based epigenome editing strategy to target the PWS-ICR did not fully reactivate all of the PWS-associated PEGs, including *NDN* and *MKRN3*. Future studies using an additional set of gRNAs or different CRISPR-based editing tools are needed for the complete recovery of PEG expression from the maternal allele. In addition, *UBE3A* silencing by *UBE3A-ATS* activation might adversely induce the pathology of Angelman syndrome. For mUPD patients, controlling monoallelic versus biallelic editing is currently challenging, as both alleles share identical sequences and dCas9 is unable to distinguish them. Further research is required to effectively manage monoallelic versus biallelic editing. Since epigenetic drift can occur during long-term in vitro culture and with aging in vivo[26], future studies are important to determine whether CRISPR-mediated epigenome editing provides stable imprinting control in the long term. To facilitate clinical applications, it is important to evaluate the methods for delivering epigenome editing and their safety. Lentivirus and plasmid lipofection may have limitations in translational applications. This challenge might be overcome by utilizing AAV in conjunction with smaller DNA-binding proteins, like zinc finger proteins[27]. Additionally, employing tissue-specific enhancers and/or engineering AAV capsid could facilitate hypothalamus-specific delivery. To minimize potential off-target effects, a self-silencing strategy[27] presents a promising method.

## Methods
### Human stem cell culture
Three human control iPSC lines (201B7, TIG119, and WD39) and one PWS line (2PWS8) were established in previous studies[10,28–30]. Three PWS iPSC lines (3PWS14, 4PWS1, and 5PWS10) were newly established in this study as previously described[31]. These iPSCs were maintained in 37 °C, 5.0% CO$_2$ incubator. For epigenome editing, the cells were cultured in StemFit AK02 media (Ajinomoto) on iMatrix coating plate. Before hypothalamic organoid generation, cells were cultured in StemScale PSC Suspension Media (Thermo Fisher Scientific). Single-cell clones of epigenome-edited iPSCs used in this study are summarized in Supplementary Table 1. All experimental procedures for iPSCs derived from patients were approved by the Keio University School of Medicine Ethics Committee (approval no. 2008-0016) and by the Tokyo Medical University Ethics Committee (approval no. T2023-0094). Informed consents were obtained from all donors of cells. These newly established iPSC lines are available from the lead contact on request.

### dCas9-Suntag-TET1 plasmids
The plasmid encoding dCas9-Suntag-TET1 components was obtained from Addgene (pPlatTET-gRNA2, #82559) and modified for the experiment. The plasmid was separated into dCas9-GCN4s and scFv-sfGFP-TET1CD, and each was integrated into a transient expression vector under CAG promoter.

### Lentivirus production
Lentivirus was prepared in-house or by VectorBuilder. For in-house production, annealed oligonucleotides for five gRNAs targeting PWS-ICR (Supplementary Fig. 1e) were ligated into an in-house-made gRNA-expressing lentiviral vector under U6 promoter (one gRNA per vector). Each of 6.0 μg lentivirus vectors was transfected with 3.0 μg pCAG-HIVgp and 3.0 μg pCMV-VSV-G-RSV-Rev into HEK293T cells by polyethyleneimine in a 10-cm sub-confluent dish in DMEM with high glucose (Thermo Fisher Scientific) supplemented with 10% heat-inactivated FBS (Sigma Aldrich) and 1% penicillin/streptomycin solution (Nacalai Tesque). On the following day, the medium was replaced with DMEM with 10% FBS containing 10 μM forskolin (Wako). At 2 days after the medium change, the lentivirus-containing supernatant was collected and passed through a 0.2 μm filter. The filtrated supernatant was ultracentrifuged at 25,000 rpm for 2 h at 4 °C. The ultracentrifuged supernatant was additionally concentrated by ultrafiltration using Amicon ultra 100 K 0.5 mL filters (Millipore). The ultrafiltrated lentivirus was dispensed and stored at −80 °C. For VectorBuilder production, a single gRNA lentiviral vector with PGK-mCherry was designed for five gRNAs, and lentivirus was produced and concentrated at VectorBuilder.

### Gene transfection
Cocktail of lentivirus expressing gRNAs was infected at approximately MOI 10 on the next day, after 12,000 iPSCs were seeded to one well of a 6-well plate in StemFit AK02 containing 10 μM Y-27632. On the next passage, 20,000–200,000 iPSCs were again seeded to one well of a 6-well plate in StemFit AK02 containing Y-27632. On the next day, gene transfection was performed according to the manufacturer's protocol of the Lipofectamine Stem reagent (Thermo Fisher Scientific). In brief, 1.0 μg CAG-dCas9-GCN4s and 1.0 μg CAG-scFv-sfGFP-TET1CD plasmids in the 100 μL Opti-MEM (Thermo Fisher Scientific) were added to 100 μL Opti-MEM with 8 μL Lipofectamine Stem reagent and incubated at room temperature for 10 min. After incubation, mixed reagents were dropped on seeded cells with StemFit AK02 containing Y-27632.

### RT-qPCR
RNA was isolated using an RNeasy Mini Kit (QIAGEN) according to the manufacturer's protocol with DNase I treatment. Total RNA (1.0 μg) was reverse-transcribed in ReveTra Ace qPCR RT master mix (Toyobo). The resultant cDNAs were diluted to 5 ng/μL in nuclease-free water. Quantitative PCR was performed using TB Green Premix ExTaq II (TaKaRa Bio) on the ViiA 7 Real-Time PCR System (Thermo Fisher Scientific) according to the manufacturer's instructions. Data were analyzed using the comparative (ΔΔCt) method. Values were normalized to *ACTB* levels and presented as relative expression levels compared to control conditions. The primers are listed in Supplementary Table 2.

### Methylation qPCR
Genomic DNA was isolated with the DNeasy Blood & Tissue kit (Qiagen). EpiScope methylated and unmethylated HCT116 gDNA (TaKaRa Bio) were used as positive and negative controls. For measuring the methylation fractions in the PWS-ICR, 200–500 ng of genomic DNA was digested with (i) 10U HpaII (New England BioLabs) and 5U MseI (New England BioLabs), (ii) 20U MspI (New England BioLabs) and 5U MseI, and (iii) 5U MseI for 12 h at 37 °C, followed by heat inactivation at 80 °C for 20 min. The reactions were 5× diluted with water, and qPCR

analysis to amplify the PWS-ICR region was performed with TB Green Premix Ex Taq GC (TaKaRa Bio) on a ViiA 7 Real-Time PCR System. A standard curve was generated, and the equation (Supplementary Fig. 3e) for the trend line was used to determine the methylation fractions. The methylation fractions were calculated as the amplification levels of HpaII + MseI-treated samples relative to those of MseI-treated samples. MspI + MseI-treated samples were used as controls to check the complete genomic digestion by these enzymes.

For measuring the methylation of *MAGEL2*, genomic DNA was digested with (i) 10U HpaII and 5U AluI (New England BioLabs), and (ii) 5U AluI. For measuring *MKRN3* methylation, genomic DNA was digested with (i) 10U HinP1I (New England BioLabs) and 5U MseI, and (ii) 5U MseI. For measuring the methylation in off-target candidates #1 and 3, genomic DNA was digested with (i) 10U HpyCH4IV (New England BioLabs) and 5U MseI, and (ii) 5U MseI. For off-target #2, genomic DNA was digested with (i) 10U HpaII and 5U MseI, and (ii) 5U MseI. qPCR analysis was performed in the same way as in the PWS-ICR methylation qPCR. The primers are listed in Supplementary Table 3.

### Identification of potential off-target sites

We searched for potential off-target sites using CRISPRdirect[11]. For each gRNA, 12 and 8 bases in the 3′ region of the target sequence adjacent to the PAM were searched against the GRCh38 genome. We tested the methylation status of adjacent CpG sites by methylation qPCR and whole-genome nanopore sequencing.

### Bisulfite sequencing

Bisulfite conversion was performed using EZ DNA Methylation Gold Kit (Zymo). The PWS-ICR was amplified using EpiTaq HS (TaKaRa Bio). Primers used are as follows: forward 5′- TAGAATAAAGGATTTTAGGGTT-3′, reverse 5′-TACTTATAATTTCTAAAAACCCCC-3′. PCR products were cloned into the pCR2.1-TOPO vector (Thermo Fisher Scientific), and the sequence of each clone was determined by Sanger sequencing. Sequence data were aligned and analyzed using QUMA[32].

### Nanopore long-read sequencing

Genomic DNA samples from iPSCs and organoids were subjected to targeted sequencing using the nanopore adaptive sampling technology as previously described[33]. Libraries were loaded onto an R9.4.1 or R10.4.1 flow cell for sequencing on a GridION instrument (ONT). CpG methylation call was performed using Guppy and Megalodon. Data were visualized by Methylartist[34].

Whole-genome nanopore sequence was performed with gRNAs(+) and gRNAs(−) iPSCs. Libraries were loaded onto an R10.4.1 flow cell, and CpG methylation call was performed using Guppy. Modkit (v0.4.2) was used to generate bedMethyl files from modified BAM, followed by global methylation comparison by MethylKit[35] (v1.24.0). The bases that are not covered in all samples are discarded by "unite." For potential off-target effect analysis, methylation information over 1000 bp tiling windows was summarized by "tileMethylCounts" (win.size = 1000, step.size = 1000, cov.bases = 5). For characterizing methylation difference, methylation information over 100 bp tiling windows was summarized, and percent methylation difference was exported as a bedGraph and visualized in IGV[36] (v2.16.0).

### Sanger sequence of PWS-ICR

The PWS-ICR was amplified from genomic DNA of iPSCs using Primer Star MAX (Takara Bio) with the following primers: forward 5′-GACCTGAGGGTGAGTGTAAATTAG-3′, reverse 5′-CACTATTATACACC TACCTGCGCTC-3′. PCR products were purified with QIAquick PCR purification kit (Qiagen) and sequenced with the forward primer.

### Short tandem repeat analysis

Short tandem repeat analysis was performed by TaKaRa Bio Inc. using the Promega Gene Print 10 system multiplex PCR to amplify 10 loci

(D21S11, TH01, TPOX, vWA, AMEL, CSF1PO, D16S539, D7S820, D13S317, and D5S818).

### Bulk RNA sequencing

RNA was isolated using an RNeasy Mini Kit (QIAGEN) according to the manufacturer's protocol with DNase I treatment. The indexed cDNA libraries were prepared using TruSeq stranded Total RNA with Ribo Zero Plus (Illumina) and sequenced using a NovaSeqX (Illumina) to obtain 150-bp paired-end reads at Macrogen. Raw fastq files were trimmed to remove low-quality bases and adapters using fastp[37] (v0.23.2). HISAT2[38] (v2.2.1) was used to map sequencing reads to the human GRCh38 genome. Coverage tracks were visualized with IGV[36] (v2.16.0). Counts were calculated using featureCounts[39] (v2.0.1) and normalized by variance-stabilizing transformation using DESeq2[40] (v1.38.3).

### Hypothalamic organoid generation

To generate hypothalamic organoids, we used a serum-free floating culture of embryoid body-like aggregates with quick aggregation (SFEBq)[41,42]. On day 0, iPSCs were dissociated into single cells with TrypLE Select and 20,000 cells per well were reaggregated in low-cell-adhesion 96-well plates with V-bottomed conical wells (Sumitomo Bakelite) in the 100 μL growth factor-free Chemically Defined Medium (gfCDM), containing IMEM + GlutaMAX (Thermo Fisher Scientific)/F-12 + GlutaMAX (Thermo Fisher Scientific) (1:1), 5 mg/mL BSA (Wako), 1% CDLC (Thermo Fisher Scientific), 1-Thioglycerol (Sigma Aldrich) with 10% KnockOut Serum Replacement (KSR; Thermo Fisher Scientific) and 20 μM Y-27632. On day 3, 100 μL gfCDM + 10% KSR without Y-27632 was added to each well. On days 6–15, gfCDM + 10% KSR with 2 μM Smoothened agonist (SAG; Cayman Chemical) was half changed every 3 days. On day 21, aggregates were collected and transferred to EZ sphere dish (IWAKI) and cultured at a 40% oxygen concentration. gfCDM + 10% KSR with 2 μM SAG was changed every 3 days. From day 51, aggregates were cultured in gfCDM + 20% KSR with 2 μM SAG, and the medium was changed every 3 days. The aggregates were collected and analyzed at days 30, 60, or 100.

### Organoid fixation and immunohistochemistry

The organoids were fixed in 4% paraformaldehyde overnight, followed by PBS wash for 5 min three times. The organoids were cryoprotected in 30% sucrose solution and embedded in OCT compound (Sakura). The organoids were cryosectioned, and the slice were put on the slide glass. The sectioned samples were 105 °C 10 min autoclaved in Target Retrieval Solution (Dako), and incubated in 0.3% Triton X-100 for 10 min following the incubation with 5% BSA blocking buffer for 1 h at room temperature. Sections were incubated overnight at 4 °C with primary antibodies at the following dilutions: RAX (guinea pig, TaKaRa Bio, M229, 1:500), NKX2-1 (mouse, Thermo Fisher Scientific, MA5-13961, 1:50), OTP (rabbit, GeneTex, GTX119601, 1:500). The samples were again washed three times with PBS and incubated with secondary antibodies conjugated with Alexa Fluor 488, Alexa Fluor 555, or Alexa Fluor 647 (Life Technologies) and Hoechst33342 (Dojindo Laboratories) for 1 h at room temperature. After washing three times with PBS, the preparation was mounted with PermaFluor Aqueous Mounting Medium (Epredia) and examined by using an LSM-710 confocal laser-scanning microscope (Carl Zeiss).

### Single-cell RNA sequencing

Four hypothalamic organoids at d63–66 from one differentiation were dissociated into a single-cell suspension using Papain Dissociation System kit (Worthington Biochemical). The dissociated cells were resuspended in ice-cold PBS containing 0.04% BSA (Miltenyi Biotec). 10,000 cells were loaded onto a Chromium Single Cell 3′ Chip (10x Genomics) and processed through the Chromium controller to generate single-cell gel beads in emulsion. The libraries were prepared

with the Chromium Single Cell 3′ Library & Gel Bead Kit v3 (10x Genomics), and sequenced on a DNBSEQ instrument (BGI) at Genewiz.

Gene expression count matrices were generated using Cellranger (v7.0.1) with GRCh38 reference (GENCODE v32/Ensembl 98) and were analyzed using Seurat[43] (v4.3.0). We excluded cells with more than 10,000 or less than 1000 detected genes, those with less than 2000 UMI counts, and those with a mitochondrial content higher than 20%, resulting in 7918 gRNAs (−) and 8786 gRNA(+) cells passing QC. Gene expression was then normalized using a global-scaling normalization method (normalization.method = "LogNormalize," scale.factor = 10,000), and the 2000 most variable genes were selected (selection.method = "vst"). gRNA(+) and (−) datasets were integrated using an anchor-based integrated strategy[44]. "FindIntegrationAnchors" and "IntegrateData" functions used the anchor object to integrate both datasets with the default parameter. The top 10 principal components were utilized for visualization with UMAP and clustering ("FindNeighbors" and "FindClusters" functions with a resolution of 0.3). Based on the expression of known markers, we grouped together clusters that were originally separate.

Count matrices of published scRNA-seq data of hypothalamus organoids[17] were downloaded via NCBI GEO (GSM4996696-7, GSM4996700-1), imported into Seurat, and normalized using a global-scaling normalization method. Our data and these published data were integrated, UMAP-visualized, and clustered as described above.

By using VoxHunt[18] (v1.0.1), we performed spatial similarity mapping of scRNA-seq data onto E15.5 mouse brains based on the ISH data from the Allen Developing Mouse Brain Atlas. The 500 most variable features from the ISH data were selected, and the similarity was calculated.

Fold change of gene expression change between gRNA(+) and (−) was calculated using "FindMarkers" function (logfc.threshold = 0, min.pct = 0.01) in Seurat. Gene set enrichment analysis[45] (GSEA) was performed using "gseGO" function (ont = "BP," minGSSize = 10, maxGSSize = 500, eps = 0, pvalueCutoff = 0.05, pAdjustMethod = "BH") in clusterProfiler[46] (v4.6.2) on all genes ranked by fold change between gRNA(+) and (−). For comparison with PWS patient data, a list of differentially expressed genes in the hypothalamus of PWS patients versus controls was used[19]. For comparison with PWS organoid data, published bulk RNA-seq data[17] were downloaded via NCBI SRA (SRR13337114-9, SRR13337126-21) and were processed with fastp[37] (v0.23.2) and Salmon[47] (v1.6.0) using the transcript index from GRCh38. The fold changes of gene expression between control and PWS organoids were calculated by DESeq2[40] (v1.38.3). Ranking all datasets based on fold change, comparisons were made using the improved rank−rank hypergeometric overlap analysis (RRHO2[20], v1.0) with Benjamini−Hochberg correction. The P-values obtained from the "FindMarkers" results can be technically used for ranking in GSEA and RRHO2; however, the inflation of P-values in "FindMarkers," which treats each cell as an independent sample, can pose challenges, particularly when comparing a single sample for each condition. Therefore, we used the fold change of gene expression as the ranking metric for these analyses.

### Calcium imaging
Hypothalamic organoids were infected with lentivirus encoding jGCaMP8s[48] under hSyn1 promoter (VectorBuilder). Infected organoids were cultured with BrainPhys Neuronal Medium (Stemcell Technologies) with B-27 Supplement (Thermo Fisher Scientific). At d100–121, organoids were transferred to glass-bottom 12-well dishes (Cellvis) and imaged using LSM-980 confocal microscope (Carl Zeiss).

For analysis, regions of interest were manually drawn, and raw fluorescent intensities were measured using Fiji[49] (v2.14.0). Raw fluorescent intensities were transformed into relative changes in fluorescence: $\Delta F/F_O = (F-F_O)/F_O$, where $F$ is the fluorescence intensity at each time point and $F_O$ is the lower fifth percentile value of the session. To quantify the activity of each cell, calcium activity events were

detected using a median absolute deviation (MAD) based method. Candidate activity events were identified by detecting a specific pattern in three consecutive time point differences—a positive slope followed by a non-positive change and then a negative slope. A candidate was retained as a true activity event only if the fluorescence signal at that time point exceeded a dynamic threshold, defined as either 2 times the MAD of the $\Delta F/F_O$ trace or a fixed minimum value of 0.1, whichever was greater. Additional steps were implemented to remove false-positive detections by examining the intervals between candidate events; closely spaced events were only retained if the fluorescence signal between them dropped below the threshold, thereby ensuring that each event represented a distinct peak. Cells with no detectable activity were excluded from the analysis. The activity rate for each cell was calculated as the number of detected activity events divided by the duration of the imaging session.

### Quantification and statistical analysis
Data were reported as the mean ± SEM. Each dot represents the value of each replicate. Statistical analyses were performed by two-tailed unpaired t-test, one-way ANOVA with Turkey correction, one-way Brown-Forsythe and Welch ANOVA with Dunnett T3 correction, and Kruskal−Wallis test with Dunn's correction. P-values < 0.05 were considered significant. For RT-qPCR data, raw $\Delta Ct$ values ($Ct_{target}$−$Ct_{ACTB}$) were used to ensure linearity and satisfy assumptions of normality for parametric testing. GraphPad Prism (v.10.0.0) was used for statistical analyses.

### Reporting summary
Further information on research design is available in the Nature Portfolio Reporting Summary linked to this article.

### Data availability
Gene expression data of scRNA-seq have been deposited in the NCBI's Gene Expression Omnibus and are accessible through the GEO Series accession number GSE262700. Source data are provided with this paper.

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

## Acknowledgements

The authors greatly thank Satoru Morimoto, Fumiko Ozawa, Shiho Nakamura (Keio University), and Mitsuru Ishikawa (Fujita Health University) for helping functional analysis, Sho Yoshimatsu (Keio University) for advice on plasmid construction, Shinya Yamanaka (Kyoto University) for 201B7 and TIG119 iPSCs, Takashi Sasaki (Keio University) for transcriptomic analysis, Naoko Yamamoto (Tokyo Medical University), Miki Sato, and Rei Murakami (Keio University) for technical support, and all the members of the Okano laboratory for encouragement and generous support for this study. This study was supported by funding from the Japan Agency for Medical Research and Development (AMED) (Grant Number 22bm0804003h0006 to H. Okano; Grant Number 23bm14230009h0001 to H. Okuno; Grant Number JP24ek0109760 to K.K.; Grant Number JP24ek0109672 to K.K. and F.M.) and Japan Society for the Promotion of Science (JSPS) (Grant Number 22J11876) to A.N.

## Author contributions

A.N., K.I., H. Okuno, and H. Okano conceived the study. A.N. and Y.H. performed the cell culture experiments. A.N., K.I., and H.I. performed the molecular biology experiments. A.N., K.I., and H.I. performed the transcriptomics analysis. A.N., K.I., F.M., M.Y., and K.K. performed the nanopore long-read sequence analysis. S.S. provided PWS patient-derived cells. A.N., K.I., H. Okuno, and H. Okano interpreted the data and wrote the manuscript. All authors have read and agreed to the published version of the manuscript.

## Competing interests

H. Okano is a paid scientific advisory board member of SanBio, Ltd. and K Pharma, Inc., but these companies had no control over the interpretation, writing, or publication of this study. All authors declare neither financial nor non-financial conflict of interest with regard to this study.
