## [Peer Review file · Nature Communications]

Rescue of imprinted genes by epigenome editing in human cellular models of Prader-Willi syndrome

Corresponding Author: Professor Hideyuki Okano

Version 0:

Reviewer comments:

Reviewer #1

(Remarks to the Author)

In this manuscript the author tackle to hypothesis that imprinted genes on the maternal chromosome can be demethylated using epigenomic editing and rescues the phenotype of Prader-Willi syndrome. Prader-Willi syndrome occurs when there is a loss of paternally expressed genes on chromosome 15q11-13, which results in hypothalamic dysfunction. Paternally expressed genes can be lost by deletions, maternal uniparental disomy, and imprinting defects. However, an individual still has copies of these genes on their maternal chromosome, but they are epigenetically silenced. Nemoto et al., use a CRISPR/dCas9-Suntag-TET1 system to target the PWS imprinting control region on the maternal chromosome in iPSCs derived from PWS patients. The main findings are 1) they are able to achieve highly efficient demethylation of the PWS ICR region and concomitant activation of PWS loci genes in iPSCs 2) this epigenetic rescue is maintained during differentiation to hypothalamic organoids from these iPSC lines, and 3) evaluate the phenotypic rescue in hypothalamic organoids by single cell RNAseq compare controls. Overall, this is a rigorous and novel use of cutting-edge technologies to better understand epigenetic dysregulation and provides important proof of principle preclinical data for the utility of these therapeutics.

Major issues:

1. More details about the editing strategy: Add a sentence or 2 more in the results that details the editing. More information is also needed in the methods. Is the gRNA lentiviral vector a single vector with the gRNAs multiplexed together or is it a single gRNA per vector? What was the MOI? What was the estimated efficiency of lipofection and was there any selection performed? Are there any GFP images available? Why was the dCAS9-Suntag-TET plasmid split into 2 separate plasmids since both proteins are required to be in the same cell for editing to work and this decreases the chances of getting 2 plasmids in the same cell. Sup Fig1g: add text to explain GCN4s is sun tag. Modify figure S1h to include more details.
2. For all results, more details specifying when, how many and which of the clonal edited iPSC lines were used and the variation observed amongst the different clones. This should be added to the text in the results and also a supplemental table specifying the clones/info for each figure/result. For the organoid single cell RNAseq, how many organoids were used per sample, how many biological replicates were performed per control, PWS and PWS-edited sample? How many cells were sequenced in total and that passed QC, and how many cells per sample?
3. Differential expression analysis of scRNAseq: the authors use the "find markers" function which counts each cell as a replicate and overly inflates the p value. The authors should perform pseudobulking and then differential expression analysis. For more details see the Bioconductor vignette: <https://bioconductor.org/packages/release/bioc/vignettes/glmGamPoi/inst/doc/pseudobulk.html>. It is possible that no FDR significant genes are found after pseudobulking. If that happens, the authors could add report this result in the results and the move to the less rigorous results obtained previously.
4. More experiments are needed to understand the mechanism of the epigenomic editing stability during differentiation. Given the relatively few clones that demonstrated activation and the need for clonal selection two mechanisms may occur: 1) either demethylation of the ICR in iPSCs occurs by the transient expression of dCAS9-Tet and this demethylated state is inherited over subsequent cell divisions and differentiation; or 2) the dCAS9-Tet expression is maintained during the cell division and differentiations and needed to maintain the demethylated status. While I believe the first possibility is likely what is happening, the authors need to explore possibility 2 by assaying for dCAS9 and Tet expression in the differentiated organoids.

5. Thinking about this as a therapeutic, can the authors add more to the discussion on how this could be used as a therapeutic and what would be needed to improve it. When/what cells would this work in and the limitations that it wasn't applied to post-mitotic neurons, what would they need improve efficiency of the CRISPR system?

Minor issues

1. Extended figure 1a – label which is paternal/maternal chromosome.
2. 3a – show me how UBE3A-ATS and UBE3A interact
3. In Extended Data Figure 3g, there were two clones of the mUPD that were fully demethylated. I would like the authors to discuss the risk of this applying this therapy to a case that is mUPD, not a deletion. How do we prevent biallelic demethylation?
4. Why was scRNA-seq of the organoids performed at day 63-66 of differentiation?
5. Line 266: "UBE3A silencing by UBE3A-ATS activation might inadequately induce the pathology of Angelman syndrome.." Replace Inadequately with adversely.
6. P values are missing from most figures and legends.

Reviewer #3

(Remarks to the Author)

In the work of Akisa Nemoto, Kent Imaizumi, and colleagues entitled "Rescue of imprinted genes by epigenome editing in human cellular models of Prader-Willi syndrome," the research aims to address the therapeutic potential of epigenome editing for Prader-Willi syndrome (PWS). PWS is a genomic imprinting disorder characterized by the loss of function of paternally expressed genes (PEGs) on chromosome 15q11-13. The study leverages a CRISPR-based epigenome editing, namely CRISPR/dCas9-Suntag- TET1 system to demethylate the PWS imprinting control region (PWS-ICR) in induced pluripotent stem cells (iPSCs) derived from PWS patients.

The main results show that successful demethylation of the PWS-ICR restored PEG expression from the maternal allele and reorganized methylation patterns in other PWS-associated imprinted regions. This reactivation persisted in selected iPSC clones during differentiation into hypothalamic organoids, leading to a reversal of gene expression patterns associated with PWS pathology. Lastly, the the authors draw a conclusion that 'by using an iPSC-based human cellular model of PWS, that their study provides the first proof of principle for CRISPR-mediated epigenome editing to treat PWS.

While this work provides some new insights into the druggability of the PWS genomic locus using CRISPR-mediated epigenome editing, it primarily serves as a weak proof of principle, and that is unfortunately my major concern. The study demonstrates that targeting the PWS locus for CRISPR-dependent reactivation in iPSCs leads to substantial, but incomplete corrections in the transcriptome of 'PWS-corrected' iPSC-derived hypothalamic organoids. However, the experimental approach undertaken by the authors lacks translatability to real-life PWS cases, even when utilizing patient-derived iPSCs. It has been widely reported that pathophysiology of PWS models occurs early during development and these traits can be identified when utilizing in vitro 3D modeling using brain organoids. However, medical intervention, whether it would be by utilizing CRISPR-dependent editing or other modalities would occur on the neuronal and not pluripotent level. To strengthen the proof of concept, it would be more impactful to target the PWS-ICR for reactivation after differentiation into post-mitotic neurons. By reactivating the PWS locus and PEGs in patient-derived iPSCs rather than in post-mitotic neurons, it is not surprising to observe sufficient rescue during and upon differentiation.

Moreover, reactivation of PWS locus has already been proven to lift some of the PWS-associated neuronal phenotypes, therefore the only novelty here is that it can be done with targeted epi-editing. To my opinion it is not sufficient to demonstrate that this widely characterised 'reactivation' attribute of CRISPR epi-editors can also be applied to PWS locus.

Moreover, this study is lacking any functional data. Electrophysiological recordings from neurons derived from individuals with Prader-Willi Syndrome (PWS) reveal altered spontaneous activity and network functionality, suggesting disrupted neurophysiological processes in PWS. The delayed maturation of spontaneous activity, along with increased susceptibility to drug modulation, further underscores these disruptions. The manuscript neither addresses neuronal patho-activity in PWS neurons nor its reversal upon epi-editing.

To sum up, I do not recommend this manuscript to be published at Nature Communications.

Minor comments:

Potential off-target effects: Authors emphasize importance of identifying potential off-target effects when utilizing CRISPR epi-editing. However, in my humble opinion the authors fall short when it comes to addressing this potential safety issue by demonstrating data for three potential candidates (Extended Data Fig. 4). To fully address potential for off-targets, I suggest performing in-silico analysis for potential genomic target sequences with homology to sgRNA seed sequence (PAM+12Ns) followed by expression profiling (RNA-seq) and cross-correlation.

Reviewer #4

(Remarks to the Author)

Nemoto et al reported the findings of CRISPR-based epigenome editing to modulate the DNA methylation status of the PWS imprinting center (PWS-ICR) in induced pluripotent stem cells (iPSCs) derived from PWS patients. They used Suntag to enhance the possibility of sgRNA-binding target sites for demethylation on CpGs. Authors applied the tool into iPSC derived hypothalamic neurons from PWS patients with different genotypes including deletions and UPD to unsilence the imprinted gene expression primarily for SNRPN from maternal chromosome 15q11-q13. They showed the efficacy on the case with a large deletion of PWS mainly. Overall, compared to the earlier reports (Liu et al., 2016, Cell 167, 233–247; Nunez et al., 2021, Cell 184, 2503–2519), the efficacy is enhanced. The study was well executed but experimental design could be improved. The increased efficacy is meaningful for translational application to human but the lenti-virus mediated delivery limits its translational application in human. There are a few intriguing and unexpected observations such as unsilencing MAGEL2 but not NDN. More investigation to understand the mechanism may be desirable since this is an important question to understand how does PWS-IC regulate the imprinted expression of many genes in 15q11-q13 domain. Overall, it is an interesting study but additional studies may be needed to strengthen the conclusion and support the future clinical studies

Major comments

- For the off target event assessment, no description how the 4 candidates were nominated. Also, the assessment is not comprehensive. Genome wide DNA methylation methods are available and well established. It seemed that it may be desirable and better method to assess the 15q11-q13 region and genome wide.
- It was stated that “The expression of SNORD116 and IPW, both of which are located in the intron of the SNRPN gene.....”. The SNORD116 is processed from the intron of SNHG14 and IPW is part of SNHG14 long non-coding transcript. Additional primer pairs across of SNHG14 gene may be necessarily
- The reactivation MAGEL2 but not NDN was interesting and intriguing too. NDN loci is closer to PWS-ICR than MAGEL2. Could this due to the off target event of dCas9-Tet1? How to approve that this is not? The secondary demethylation was proposed. For the assessment of future therapeutic potential for this approach, it may be essential and valuable to perform additional study to understand the mechanism or exclude some possibilities. Further investigation is also desirable since this is an important mechanistic question for the role of imprinting center in 15q11-q13 domain.
- One of the important questions for dCas9 epigenome editing in translational application is the duration of editing effect? It may be desirable to show how long does the demethylation effect in non-dividing neurons or dividing cells last after the single administration
- In Figure 1D, methylation status in PWS iPSC representing the maternal chromosome, is about 75%-120% or 150%. Could author explain how the level could be >100%
- In Figure 1G, following the authors’ description and also true in literature, the SNORD116 and IPW are part of long noncoding SNHG14 transcript. However, in the data presented, the expression level of SNORD116 is lower than IPW/SNHG14. That is unexpected since the SNORD116 has many copies and is processed RNAs from IPW or SNHG14. Also, a negative control with RT- for SNORD116 is necessary after the DNAase treatment of RNA preparation since SNORD116 is intronless
- 4) MKRN3 promoter region in organoid didn’t maintain demethylation (Fig 3E). However, MKRN3 expression was unsilenced and increased expression from maternal chromosome. This result may suggest dCas9-Tet1 is not related to MKRN3 reactivation and qPCR results may reflect DNA contamination. Or other mechanistic explanation that is important to understand the role of PWS-IC
- As a positive control to compare the efficacy, authors may consider the DNMT inhibitor (ex;5-Aza) to see PEG expression from maternal chromosome in PWS iPSCs?

Reviewer #5

(Remarks to the Author)

Nemoto et al has generated enhanced version of dCas9-Tet1 as an epigenome editing tool. They used Suntag to provide multiple platform to increase the possibility of gRNA-binding target sites for demethylation on CpGs. They applied the tool into iPSC derived from PWS patients with different types of deletions, UPD, and imprinting defect to unsilence the imprinted gene expression from maternal chromosome. They could show efficacy on large deletion of PWS mainly, rather than UPD. Overall, compared to the original ideas (Liu et al., 2016, Cell 167, 233–247; Nunez et al., 2021, Cell 184, 2503–2519), it seems like that its efficacy is enhanced but the edited single-cell derived clones and organoid definitely showed almost 100% rescued expression. I’d like to suggest that increased efficacy by initial transduction is meaningful for application to human, even if it is lenti-vial mediated delivery (lenti-virus is not appropriate to human, though). To prove the concept of principle as they suggest, the following questions should be addressed.

My major concerns are,

1) Considering the impact of dCas9-Tet1 with 5 guide RNA on wide range, (as my knowledge as well as authors mentioned, its covering target site for epigenome editing is 200 bp), demethylation on multiple loci, especially only the promoter region of each gene (SNRPN, MAGEL2, MKRN3) was shown in iPSCs. how could they recognize CpG sites on promoter only, not the whole CpG sites near the PWS-ICR? Could they provide methylation status profiling of MKRN3-SNORD115 region?

2) Figure 1D, methylation status in PWS iPSC representing the maternal chromosome, is about 75%-120% , 150%. Could

you explain the meaning of these variable value?

3) in Figure 1G, following the authors' mentions, if the SNORD116 and IPW is part of SNRPN transcript (exactly, I would correct it as the part of SNHG14), the expression level of SNORD116 and IPW could be similar. In case of SNORD116, the variation of the each replicate is too broad range to argue the rescue the loss of gene expression.

4) MKRN3 promoter region in organoid didn't maintain demethylation (Fig 3E). However, MKRN3 expression was unsilenced and increased expression from maternal chromosome. This result may suggest dCas9-Tet1 is not related to MKRN3 reactivation and qPCR results may reflect DNA contamination.

5) As a positive control, have the authors been tried to use DNMT inhibitor (ex;5-Aza) to see PEG expression from maternal chromosome in PWS iPSCs ?

6) NDN loci is closer to PWS-ICR than MAGEL2. However, MAGEL2 was only unsilenced, not NDN. Authors suggest that MAGEL2 expression is secondary demethylation governed by PWS-ICR. They may need solid mechanistic evidences to insist this model.

Version 1:

Reviewer comments:

Reviewer #1

(Remarks to the Author)

The authors have satisfactorily addressed my concerns. I support acceptance.

Reviewer #3

(Remarks to the Author)

Reviewer #5

(Remarks to the Author)

Point-by-Point Responses to Reviewers

We appreciate the reviewers' constructive comments and suggestions to improve our manuscript. We are delighted to see that the reviewers thought that 'this is a rigorous and novel use of cutting-edge technologies to better understand epigenetic dysregulation', 'this work provides some new insights into the druggability of the PWS genomic locus using CRISPR-mediated epigenome editing', and they considered that, overall, this study 'provides important proof of principle preclinical data for the utility of these therapeutics.' To address their concerns, we now performed several new experiments and revised the manuscript along the lines summarized below and detailed in the point-by-point response in this letter. We believe that our manuscript has been significantly strengthened by this revision.

In these point-by-point responses, the original reviewer comments are shown in **black**, while our responses are shown in **blue**. In brief, here are the major new experiments and analyses.

1. We have now expanded our **off-target analysis** more comprehensively. We performed the whole-genome DNA methylation profiling, and found that the epigenome editing did not alter the DNA methylation status of potential off-target genomic regions that share homology with the gRNA sequences. Furthermore, RNA-seq analysis confirmed that the expression of these potential off-target genes remained unchanged following epigenome editing.

2. We addressed **the mechanism of *MAGEL2* reactivation**. While our epigenome editing targeted the PWS-ICR and *SNRPN* gene, it also unsilenced another PWS-associated gene, *MAGEL2*. By closely examining the methylation dynamics during the epigenome editing process, we demonstrated that the unsilencing of *MAGEL2* is not a result of off-target effects. Our findings indicate that the epigenetic status of PWS-ICR regulates the locus-specific methylation of *MAGEL2*.

3. As a proof-of-concept for clinical intervention, we performed **the epigenome editing in differentiated organoids**. We successfully transduced dCas9-Suntag-TET1 and gRNAs with PWS organoids, resulting in the restoration of *SNRPN* gene expression at the organoid stage.

4. We assessed **the functionality of epigenome-edited organoids** by live calcium imaging, and revealed that epigenome editing reverted a reduced spontaneous activity in PWS organoids.

5. We **validated our RT-qPCR results** by checking RT- controls and bulk RNA-seq analysis.

In this revision, we added **18** new figure panels and updated **13** existing figure panels.

We hope that the newly added results and the text revision make our manuscript suitable for publication in *Nature Communications*.

Reviewer #1

In this manuscript the author tackle to hypothesis that imprinted genes on the maternal chromosome can be demethylated using epigenomic editing and rescues the phenotype of Prader-Willi syndrome. Prader-Willi syndrome occurs when there is a loss of paternally expressed genes on chromosome 15q11-13, which results in hypothalamic dysfunction. Paternally expressed genes can be lost by deletions, maternal uniparental disomy, and imprinting defects. However, an individual still has copies of these genes on their maternal chromosome, but they are epigenetically silenced. Nemoto et al., use a CRISPR/dCas9-Suntag-TET1 system to target the PWS imprinting control region on the maternal chromosome in iPSCs derived from PWS patients. The main findings are 1) they are able to achieve highly efficient demethylation of the PWS ICR region and concomitant activation of PWS loci genes in iPSCs 2) this epigenetic rescue is maintained during differentiation to hypothalamic organoids from these iPSC lines, and 3) evaluate the phenotypic rescue in hypothalamic organoids by single cell RNAseq compare controls. Overall, this is a rigorous and novel use of cutting-edge technologies to better understand epigenetic dysregulation and provides important proof of principle preclinical data for the utility of these therapeutics.

(Response) We appreciate the reviewers' constructive comments. Below we are providing details and describe a series of new experiments to address these issues:

Major issues:

1. More details about the editing strategy: Add a sentence or 2 more in the results that details the editing. More information is also needed in the methods. Is the gRNA lentiviral vector a single vector with the gRNAs multiplexed together or is it a single gRNA per vector? What was the MOI? What was the estimated efficiency of lipofection and was there any selection performed? Are there any GFP images available? Why was the dCAS9-Suntag-TET plasmid split into 2 separate plasmids since both proteins are required to be in the same cell for editing to work and this decreases the chances of getting 2 plasmids in the same cell. Sup Fig1g: add text to explain GCN4s is sun tag. Modify figure S1h to include more details.

(Response) We used lentiviral vectors containing individual gRNAs (one gRNA per vector) at approximately MOI 10. The lipofection efficiency was ~60% based on sfGFP expression (Extended Data Fig 2a), and we did not use antibiotics for selection. We tried a vector with two components of dCas9-Suntag-TET1 (dCas9-GCN4s and scFv-sfGFP-TET1CD) with 2A peptide linker; however, the editing efficiency was not as high as separate vectors. We assume that a substantial amount of uncleaved fusion proteins might hinder the efficient editing.

We added Extended Data Fig 2a for sfGFP⁺ transfected cell images, updated Extended Data Figure 1g and 1h as the reviewer suggested, and included the following sentences in the Result and Methods sections to describe the details of the editing strategy:

Extended Data Fig. 2 Characterization of epigenome-edited iPSCs

a, Representative image of iPSCs transfected with dCas9-Suntag-TET1 components (dCas9-GCN4s and scFv-sfGFP-TET1CD). Scale bar, 200 μ m.

Extended Data Fig. 1 PWS iPSC lines and strategy of epigenome editing

g, Transient overexpressing vectors for dCas9-Suntag-TET1 components. h, Schedule of epigenome editing of iPSCs.

(Page 4, Line 82)

Five guide RNAs (gRNAs) targeting the PWS-ICR (Extended Data Fig. 1e, f) were delivered to these iPSCs by lentivirus, and plasmids encoding dCas9-Suntag-TET1 components (dCas9-GCN4s and scFv-sfGFP-TET1CD) were transfected by lipofection (Extended Data Fig. 1g, h).

(Page 11, Line 329)

For in-house production, annealed oligonucleotides for five gRNAs targeting PWS-ICR (Extended Data Fig. 1e) were ligated into an in-house-made gRNA expressing lentiviral vector under U6 promoter (one gRNA per vector).

2. For all results, more details specifying when, how many and which of the clonal edited iPSC lines were used and the variation observed amongst the different clones. This should be added to the text in the results and also a supplemental table specifying the clones/info for each figure/result. For the organoid single cell RNAseq, how many organoids were used per sample, how many biological replicates were performed per control, PWS and PWS-edited sample? How many cells were sequenced in total and that passed QC, and how many cells per sample?

(Response) We thank the reviewer for this important comment. In most experiments, we used at least three independently derived clones of epigenome-edited iPSCs to ensure the robustness and reproducibility of the results. Detailed information regarding iPSC clones used for each assay has been added to the Results section and is also summarized in Supplementary Table 1 for clarity.

For scRNA-seq, due to constraints related to technical resources and instrument availability, we were only able to perform single-cell RNA sequencing on one sample from each condition (i.e. gRNAs(-) and gRNA(+)). We used four organoids from one differentiation, processed 10,000 cells for each condition, and obtained 7,918 gRNAs(-) and 8,786 gRNA(+) cells passing QC. We included this sample information and the details of the cell number, QC results, and analysis pipelines in the Method section, written in Page 15, Line 475.

3. Differential expression analysis of scRNAseq: the authors use the “find markers” function which counts each cell as a replicate and overly inflates the p value. The authors should perform pseudobulking and then differential expression analysis. For more details see the Bioconductor vignette:

<https://bioconductor.org/packages/release/bioc/vignettes/glmGamPoi/inst/doc/pseudobulk.html>. It is possible that no FDR significant genes are found after pseudobulking. If that happens, the authors could add report this result in the results and the move to the less rigorous results obtained previously.

(Response) We could only conduct single-cell RNA sequencing on a single sample from each condition, as described above. As a result, pseudobulking was not a viable option for our dataset. As the reviewer pointed out, the ‘FindMarkers’ function in Seurat treats each cell as an independent sample, which may lead to p-values that are not reliable indicators. Therefore, we employed Gene Set Enrichment Analysis (GSEA) and Rank-Rank Hypergeometric Overlap (RRHO), both of which focus on fold changes rather than p-values. To address this technical limitation, we have added the following clarification:

(Page 16, Line 516)

The P-values obtained from the 'FindMarkers' results can be technically used for ranking in GSEA and RRHO2; however, the inflation of P-values in 'FindMarkers', which treats each cell as an independent sample, can pose challenges, particularly when comparing a single sample for each condition. Therefore, we used the fold change of gene expression as the ranking metric for these analyses.

We would like to highlight that our scRNA-seq results are consistent with our functional analysis data. In our calcium imaging studies, we observed that epigenome editing restored the reduced spontaneous activity in PWS organoids (Extended Data Figure 11a–c). This finding aligns with the alterations in synapse-related genes in PWS cells, where epigenome editing was able to partially reverse these effects.

4. More experiments are needed to understand the mechanism of the epigenomic editing stability during differentiation. Given the relatively few clones that demonstrated activation and the need for clonal selection two mechanisms may occur: 1) either demethylation of the ICR in iPSCs occurs by the transient expression of dCAS9-Tet and this demethylated state is inherited over subsequent cell divisions and differentiation; or 2) the dCAS9-Tet expression is maintained during the cell division and differentiations and needed to maintain the demethylated status. While I believe the first possibility is likely what is happening, the authors need to explore possibility 2 by assaying for dCAS9 and Tet expression in the differentiated organoids.

(Response) We thank the reviewer for the insightful comment regarding the potential mechanisms underlying the stability of epigenome editing during differentiation. To address this, we checked dCas9 and sfGFP expression by RT-qPCR and found that dCas9-Suntag-TET1 was eliminated by 6 days in iPSCs after transient transduction (Extended Data Fig 2b). These results suggest that the observed demethylation was not due to sustained dCas9-Suntag-TET1 activity.

Extended Data Fig. 2 Characterization of epigenome-edited iPSCs

b, RT-qPCR analysis of the expression of dCas9 and sfGFP in iPSCs transfected with epigenome editing components.

Furthermore, we validated the absence of dCas9-Suntag-TET1 expression in differentiated organoids using RT-qPCR. Due to space constraints, we were not able to add this data of RT-qPCR analysis in organoids at day 60 to the manuscript. We are providing it below to the reviewers.

RT-qPCR analysis of the expression of dCas9 and sfGFP in organoids at d60.

5. Thinking about this as a therapeutic, can the authors add more to the discussion on how this could be used as a therapeutic and what would be needed to improve it. When/what cells would this work in and the limitations that it wasn't applied to post-mitotic neurons, what would they need improve efficiency of the CRISPR system?

(Response) We envision that epigenome editing will be utilized in postnatal brains for future clinical applications aimed at treating PWS. As a proof-of-concept for such interventions, we performed the epigenome editing in differentiated organoids. We successfully transduced dCas9-Suntag-TET1 and gRNAs following the organoid generation process, leading to the restoration of *SNRPN* gene expression in PWS organoids. These data are included in Extended Data Fig 12a–c.

Extended Data Fig. 12 Epigenome editing in organoids

a, Schedule of epigenome editing of organoids. b, Representative images of organoids transfected with mCherry-gRNAs and dCas9-Suntag-TET1 components (dCas9-GCN4s and scFv-sfGFP-TET1CD). c, RT-qPCR analysis of SNRPN expression in organoids at day 49 (gRNAs(-) dCas9-Suntag-TET1 (+), n = 3 differentiations, 1 iPSC line; gRNAs(+) dCas9-Suntag-TET1 (+), n = 3 differentiations, 1 iPSC; unpaired t-test, *P = 0.0495).

The primary limitation for clinical application is the restricted delivery of dCas9-Suntag-TET1 and gRNAs to the hypothalamus. This technical challenge could potentially be addressed using AAV with smaller-size DNA-binding proteins, such as zinc finger proteins (Neumann et al., Science, 2024). Additionally, employing tissue-specific enhancers and/or engineering AAV capsid could facilitate hypothalamus-specific delivery. Secondly, in post-mitotic cells, dCas9-Suntag-TET1 may not be efficiently eliminated after transduction, which could lead to off-target demethylation. This issue could be mitigated through self-silencing strategies (Neumann et al., Science, 2024). We have summarized these points in the Discussion section.

(Page 10, Line 300)

To facilitate clinical applications, it is important to evaluate the methods for delivering epigenome editing and their safety. Lentivirus and plasmid lipofection may have limitations in translational applications. This challenge might be overcome by utilizing AAV in conjunction with smaller DNA-binding proteins, like zinc finger proteins²⁵. Additionally, employing tissue-specific enhancers and/or engineering AAV capsid could facilitate hypothalamus-specific delivery. To minimize potential off-target effects, a self-silencing strategy²⁵ presents a promising method.

Minor issues

1. Extended figure 1a – label which is paternal/maternal chromosome.

(Response) We revised Extended Data Fig 1a as the reviewer suggested.

2. 3a – show me how UBE3A-ATS and UBE3A interact

(Response) We revised Fig 3a as the reviewer suggested.

3. In Extended Data Figure 3g, there were two clones of the mUPD that were fully demethylated. I would like the authors to discuss the risk of this applying this therapy to a case that is mUPD, not a deletion. How do we prevent biallelic demethylation?

(Response) We thank the reviewer for this important point regarding the implications of applying our epigenome editing approach to cases of maternal uniparental disomy (mUPD), where both alleles of the 15q11–13 region are of maternal origin. We included the following sentences in the Discussion part:

(Page 10, Line 297)

For mUPD patients, controlling monoallelic versus biallelic editing is currently challenging, as both alleles share identical sequences and dCas9 is unable to distinguish them. Further research is required to effectively manage monoallelic versus biallelic editing.

4. Why was scRNA-seq of the organoids performed at day 63-66 of differentiation?

(Response) Our goals of scRNA-seq analysis were to investigate the effects of epigenome editing on transcriptomic changes associated with PWS and to confirm the cell type specification into the hypothalamus. For analyzing gene expression related to PWS, it would be advantageous to utilize older samples, as we compared our data with postmortem brain data from PWS patients. However, the publicly available scRNA-seq datasets of iPSC-derived hypothalamus organoids are only available at day 40 of differentiation. Therefore, we selected days 63-66 as an intermediate stage between day 40 and the later stages. Due to our limited access to scRNA-seq instruments, we collected samples at day 63 for the epigenome-edited organoids and at day 66 for the unedited controls.

5. Line 266: “UBE3A silencing by UBE3A-ATS activation might inadequately induce the pathology of Angelman syndrome..” Replace Inadequately with adversely.

(Response) The text was revised as the reviewer suggested.

6. P values are missing from most figures and legends.

(Response) Accordingly, we performed statistical tests for each dataset and added the details in the figure legends and the Method part.

Reviewer #3

In the work of Akisa Nemoto, Kent Imaizumi, and colleagues entitled "Rescue of imprinted genes by epigenome editing in human cellular models of Prader-Willi syndrome," the research aims to address the therapeutic potential of epigenome editing for Prader-Willi syndrome (PWS). PWS is a genomic imprinting disorder characterized by the loss of function of paternally expressed genes (PEGs) on chromosome 15q11-13. The study leverages a CRISPR-based epigenome editing, namely CRISPR/dCas9-Suntag- TET1 system to demethylate the PWS imprinting control region (PWS-ICR) in induced pluripotent stem cells (iPSCs) derived from PWS patients.

The main results show that successful demethylation of the PWS-ICR restored PEG expression from the maternal allele and reorganized methylation patterns in other PWS-associated imprinted regions. This reactivation persisted in selected iPSC clones during differentiation into hypothalamic organoids, leading to a reversal of gene expression patterns associated with PWS pathology. Lastly, the authors draw a conclusion that 'by using an iPSC-based human cellular model of PWS, that their study provides the first proof of principle for CRISPR-mediated epigenome editing to treat PWS.

While this work provides some new insights into the druggability of the PWS genomic locus using CRISPR-mediated epigenome editing, it primarily serves as a weak proof of principle, and that is unfortunately my major concern. The study demonstrates that targeting the PWS locus for CRISPR-dependent reactivation in iPSCs leads to substantial, but incomplete corrections in the transcriptome of 'PWS-corrected' iPSC-derived hypothalamic organoids. However, the experimental approach undertaken by the authors lacks translatability to real-life PWS cases, even when utilizing patient-derived iPSCs. It has been widely reported that pathophysiology of PWS models occurs early during development and these traits can be identified when utilizing in vitro 3D modeling using brain organoids. However, medical intervention, whether it would be by utilizing CRISPR-dependent editing or other modalities would occur on the neuronal and not pluripotent level. To strengthen the proof of concept, it would be more impactful to target the PWS-ICR for reactivation after differentiation into post-mitotic neurons. By reactivating the PWS locus and PEGs in patient-derived iPSCs rather than in post-mitotic neurons, it is not surprising to observe sufficient rescue during and upon differentiation.

(Response) We appreciate the reviewers' constructive comments. To address this concern, we performed the epigenome editing in differentiated organoids. We successfully transduced dCas9-Suntag-TET1 and gRNAs following the organoid generation process, leading to the restoration of *SNRPN* gene expression in PWS organoids. These data are included in Extended Data Fig 12a–c.

Extended Data Fig. 12 Epigenome editing in organoids

a, Schedule of epigenome editing of organoids. b, Representative images of organoids transfected with mCherry-gRNAs and dCas9-Suntag-TET1 components (dCas9-GCN4s and scFv-sfGFP-TET1CD). c, RT-qPCR analysis of SNRPN expression in organoids at day 49 (gRNAs(-) dCas9-Suntag-TET1 (+), n = 3 differentiations, 1 iPSC line; gRNAs(+) dCas9-Suntag-TET1 (+), n = 3 differentiations, 1 iPSC; unpaired t-test, *P = 0.0495).

To discuss the current hurdles to CRISPR-mediated epigenome editing in clinical settings, we have also included the following statements:

(Page 10, Line 300)

To facilitate clinical applications, it is important to evaluate the methods for delivering epigenome editing and their safety. Lentivirus and plasmid lipofection may have limitations in translational applications. This challenge might be overcome by utilizing AAV in conjunction with smaller DNA-binding proteins, like zinc finger proteins²⁵. Additionally, employing tissue-specific enhancers and/or engineering AAV capsid could facilitate hypothalamus-specific delivery. To minimize potential off-target effects, a self-silencing strategy²⁵ presents a promising method.

Moreover, reactivation of PWS locus has already been proven to lift some of the PWS-associated neuronal phenotypes, therefore the only novelty here is that it can be done with targeted epi-editing. To my opinion it is not sufficient to demonstrate that this widely characterised 'reactivation' attribute of CRISPR epi-editors can also be applied to PWS locus.

(Response) We acknowledge that the concept of reactivation has been previously established, such as the use of epigenetic inhibitors (Kim et al., *Nature Medicine*, 2017) and histone methyltransferase gene knockdown (Cruvinel et al., *Human Molecular Genetics*, 2014); however, these methods have not achieved the full activation of PWS-associated genes to levels comparable to those in healthy individuals. In contrast, our study demonstrates that our targeted epigenome editing approach not only reactivates PWS-associated genes but does so to levels that are comparable to wild-type expression. This represents a significant advancement in the field and underscores the potential of our method as a more effective therapeutic strategy.

Additionally, these epigenetic inhibitors and histone methyltransferase gene knockdown potentially have genome-wide epigenetic remodeling effects. Our use of CRISPR technology for targeted epigenome editing allows for a high degree of specificity, minimizing the risk of unintended consequences and enhancing the safety profile of our approach.

The hierarchical regulation of imprinted genes, particularly the role of PWS-ICR demethylation in unsilencing other PWS-associated genes like *MAGEL2*, adds another layer of understanding to the complex regulatory mechanisms in the PWS locus. We believe that these findings not only advance our knowledge of PWS biology but also provide valuable insights for the development of targeted epigenetic therapeutics.

Moreover, this study is lacking any functional data. Electrophysiological recordings from neurons derived from individuals with Prader-Willi Syndrome (PWS) reveal altered spontaneous activity and network functionality, suggesting disrupted neurophysiological processes in PWS. The delayed maturation of spontaneous activity, along with increased susceptibility to drug modulation, further underscores these disruptions. The manuscript neither addresses neuronal patho-activity in PWS neurons nor its reversal upon epi-editing.

(Response) We performed live calcium imaging of organoids using jGCaMP8s, and observed decreased spontaneous activity in PWS organoids compared with control, which is consistent with previous reports. Notably, epigenome-edited organoids exhibited increased activity that was comparable to that of control organoids, suggesting the epigenome editing strategy is effective for functional restoration. These results are included in Extended Data Fig 11a–c, and we have added a corresponding description in Page 8, Line 239.

Extended Data Fig. 11 Calcium imaging of organoids

a, jGCaMP8s-expressing organoids. Scale bar, 100 μm . b, Representative heatmap of spontaneous calcium signal traces. c, Frequency of spontaneous calcium activity (WD39, $n = 267$ cells from 5 organoids, 4 differentiation, 1 iPSC line; gRNAs(-) dCas9-Suntag-TET1 (+), $n = 34$ cells from 3 organoids, 3 differentiation, 1 clone from 1 iPSC line; gRNAs(+) dCas9-Suntag-TET1 (+), $n = 133$ cells from 3 organoids, 3 differentiation, 1 clone from 1 iPSC line; Kruskal–Wallis test with Dunn’s correction, * $P = 0.0137$ [WD39 vs -/+], 0.0429 [-/+ vs +/+]; ns, not significant).

Minor comments:

Potential off-target effects: Authors emphasize importance of identifying potential off-target effects when utilizing CRISPR epi-editing. However, in my humble opinion the authors fall short when it comes to addressing this potential safety issue by demonstrating data for three potential candidates (Extended Data Fig. 4). To fully address potential for off-targets, I suggest performing in-silico analysis for potential genomic target sequences with homology to sgRNA seed sequence (PAM+12Ns) followed by expression profiling (RNA-seq) and cross-correlation.

(Response) As the reviewer suggested, we identified potential genomic regions with homology to gRNA seed sequences (PAM+12-mer and +8-mer, respectively) by the *in silico* tool CRISPRdirect. By RNA-seq, we confirmed that the expression of these potential off-target genes was not changed by epigenome editing. We added this new result in Extended Data Fig 4e.

Extended Data Fig. 4 Off-target effects of epigenome editing
e, Correlation of gene expression between gRNAs(-) and gRNAs(+) iPSCs.

Furthermore, to assess off-target effects more comprehensively, we performed the DNA methylation characterization by whole-genome nanopore sequencing. Overall, the potential off-target sites were unaffected by epigenome editing. These new analyses are included in Extended Data Fig 4a–c, and we have added a corresponding description in Page 5 Line 118.

Extended Data Fig. 4 Off-target effects of epigenome editing
a, Potential off-target sites regions with homology to gRNA seed sequence. b, Correlation of CpG methylation between gRNAs(-) and gRNAs(+) iPSCs. c, Per-read methylation status of potential off-target sites by long-read sequencing analysis. The purple highlighted region indicates the 12-mer off-target sites.

Reviewer #4

Nemoto et al reported the findings of CRISPR-based epigenome editing to modulate the DNA methylation status of the PWS imprinting center (PWS-ICR) in induced pluripotent stem cells (iPSCs) derived from PWS patients. They used Suntag to enhance the possibility of sgRNA-binding target sites for demethylation on CpGs. Authors applied the tool into iPSC derived hypothalamic neurons from PWS patients with different genotypes including deletions and UPD to unsilence the imprinted gene expression primarily for SNRPN from maternal chromosome 15q11-q13. They showed the efficacy on the case with a large deletion of PWS mainly. Overall, compared to the earlier reports (Liu et al., 2016, Cell 167, 233–247; Nunez et al., 2021, Cell 184, 2503–2519), the efficacy is enhanced. The study was well executed but experimental design could be improved. The increased efficacy is meaningful for translational application to human but the lenti-virus mediated delivery limits its translational application in human. There are a few intriguing and unexpected observations such as unsilencing MAGEL2 but not NDN. More investigation to understand the mechanism may be desirable since this is an important question to understand how does PWS-IC regulate the imprinted expression of many genes in 15q11-q13 domain. Overall, it is an interesting study but additional studies may be needed to strengthen the conclusion and support the future clinical studies

Major comments

- For the off target event assessment, no description how the 4 candidates were nominated. Also, the assessment is not comprehensive. Genome wide DNA methylation methods are available and well established. It seemed that it may be desirable and better method to assess the 15q11-q13 region and genome wide.

(Response) We thank the reviewer for pointing out the need for a more detailed explanation regarding the selection of candidate off-target sites. To address this, we have clarified the revised manuscript that we utilized the *in silico* tool CRISPRdirect to identify potential genomic regions with homology to sgRNA seed sequence (PAM+12-mer and +8-mer, respectively). We summarized this analysis in Extended Data Fig 4a:

a	20-mer match (On-target)	12-mer match	8-mer match
gRNA #1	1	3	838
gRNA #2	1	1	1,256
gRNA #3	1	0	132
gRNA #4	1	0	636
gRNA #5	1	0	726

Extended Data Fig. 4 Off-target effects of epigenome editing

a, Potential off-target sites identified by CRISPRdirect based on the sequence homology with gRNA targets.

We have also updated the Methods section to provide more information about potential off-target sites:

(Page 13, line 392)

We searched for potential off-target sites using CRISPRdirect¹¹. For each gRNA, 12 and 8 bases in the 3' region of the target sequence adjacent to the PAM were searched against the GRCh38 genome. We tested the methylation status of adjacent CpG sites by methylation qPCR and whole-genome nanopore sequencing.

As the reviewer suggested, we performed the genome-wide DNA methylation characterization using nanopore sequencing to more comprehensively assess possible off-target effects. Overall, the potential off-target sites were unaffected by epigenome editing. These new analyses are included in Extended Data Fig 4a–c, and we have added a corresponding description in Page 5 Line 118.

Extended Data Fig. 4 Off-target effects of epigenome editing
 a, Potential off-target sites identified by CRISPRdirect based on the sequence homology with gRNA targets. b, Correlation of CpG methylation between gRNAs(-) and gRNAs(+) iPSCs. c, Per-read methylation status of potential off-target sites by long-read sequencing analysis. The purple highlighted region indicates the 12-mer off-target sites.

• It was stated that “The expression of SNORD116 and IPW, both of which are located in the intron of the SNRPN gene.....”. The SNORD116 is processed from the intron of SNHG14 and IPW is part of SNHG14 long non-coding transcript. Additional primer pairs across of SNHG14 gene may be necessarily

(Response) We thank the reviewer for pointing out the inaccuracy in the original description of *SNORD116* and *IPW*. As correctly noted, *SNORD116* is processed from an intron of the *SNHG14* gene, and *IPW* is part of the long non-coding transcript of *SNHG14*, rather than originating from the intron of *SNRPN*. To address this, we updated the description of the genes in the Result section as follows:

(Page 5, line 132)

The expression of *SNORD116* and *IPW*, which are processed from the intron of the *SNHG14* gene and part of *SNHG14* transcript, respectively, was upregulated and comparable to the healthy control iPSCs (Fig. 1h, Extended Data Fig. 6a).

Additionally, we performed RNA-seq analysis and confirmed that epigenome editing upregulated the expression of PWS-associated imprinted genes across *SNHG14* region. These new analyses are included in Extended Data Fig 5, and we have added a corresponding description in Page 5 Line 131.

Extended Data Fig. 5 Transcriptomic analysis of epigenome-edited iPSCs a, Heatmap summarizing the expression of PWS-associated imprinted genes. b, RNA-seq read coverage mapped to the genomic region of *SNHG14*.

- The reactivation *MAGEL2* but not *NDN* was interesting and intriguing too. *NDN* loci is closer to PWS-ICR than *MAGEL2*. Could this due to the off target event of dCas9-Tet1? How to approve that this is not? The secondary demethylation was proposed. For the assessment of future therapeutic potential for this approach, it may be essential and valuable to perform additional study to understand the mechanism or exclude some possibilities. Further investigation is also desirable since this is an important mechanistic question for the role of imprinting center in 15q11-q13 domain.

(Response) We thank the reviewer for this insightful comment regarding the selective reactivation of *MAGEL2*. To address the mechanism of *MAGEL2* reactivation, we analyzed the methylation dynamics during the epigenome editing process. We performed gRNA lentiviral infection and dCas9-Suntag-TET1 lipofection in PWS iPSCs and then maintained the cells in bulk without single-cell cloning, allowing us to observe the temporal pattern of methylation and gene expression change right after transfection (Extended Data Fig. 7a). We observed that PWS-ICR demethylation occurred immediately after dCas9-Suntag-TET1 transfection. In contrast, *MAGEL2* demethylation was observed subsequently, particularly at least a few passages after the transfection (Extended Data Fig. 7b), by which time dCas9-Suntag-TET1 was already eliminated (Extended Data Fig. 2b). We also detected a time lag of the expression upregulation between *SNRPN* and *MAGEL2* (Extended Data Fig. 7c).

These time lags of demethylation and gene expression suggest that the demethylation of *MAGEL2* is not a consequence of off-target effects from dCas9-Suntag-TET1. Instead, it implies that the demethylation of PWS-ICR subsequently regulates the methylation status of *MAGEL2*. These new analyses are included in Extended Data Fig 7a-c, and we have added a corresponding description in Page 6 Line 152.

Extended Data Fig. 7 Methylation dynamics of PWS-ICR/*SNRPN* and *MAGEL2* during epigenome editing a, Schedule of epigenome editing of iPSCs for methylation dynamics. b, Methylation status of PWS-ICR and *MAGEL2* region in epigenome-edited PWS iPSCs measured by genomic qPCR analysis following methylation-sensitive enzyme digestion (gRNAs(-), n = 3 experiments, 1 iPSC line; gRNAs(+), n = 3 experiments, 1 iPSC line). Lines denote fitted curves by LOESS regression. c, RT-qPCR analysis of the expression of *SNRPN* and *MAGEL2* (gRNAs(-), n = 3 experiments, 1 iPSC line; gRNAs(+), n = 3 experiments, 1 iPSC line). Lines denote fitted curves by LOESS regression.

- One of the important questions for dCas9 epigenome editing in translational application is the duration of editing effect? It may be desirable to show how long does the demethylation effect in non-dividing neurons or diving cells last after the single administration

(Response) We thank the reviewer for raising this important point regarding the duration of the epigenome editing effect. The epigenome editing effect persisted for at least one month (4 passages) in iPSCs (Fig. 1c) and for 100 days in organoids (Fig. 2g). We verified that the dCas9-Suntag-TET1 was eliminated by 6 days following transient transduction (Extended Data Fig. 2b); therefore, the observed effect was not a result of prolonged dCas9-Suntag-TET1 activity, but rather the lasting impact of the single administration. These analyses are included in Extended Data Fig 2b, and we have added a corresponding description in Page 4 Line 85 and Page 6, Line 159.

Extended Data Fig. 2 Characterization of epigenome-edited iPSCs
 b, RT-qPCR analysis of the expression of dCas9 and sfGFP in iPSCs transfected with epigenome editing components.

- In Figure 1D, methylation status in PWS iPSC representing the maternal chromosome, is about 75%-120% or 150%. Could author explain how the level could be >100%

(Response) We thank the reviewer for pointing out the apparent methylation values exceeding 100% in Figure 1d. In methylation qPCR analysis, we utilized a strategy similar to $\Delta\Delta C_t$ method, which indeed resulted in larger error bars for samples with higher methylation levels. This increase in variability is primarily due to the logarithmic nature of the fold change calculations derived from the C_t values. As a result, the methylation levels may exceed 100% due to the mathematical properties of the transformation, not due to actual biological methylation levels exceeding 100%.

In our study, we employed the methylation-sensitive restriction enzyme HpaII to digest genomic DNA, allowing us to obtain C_t values for both HpaII-treated and untreated samples. The methylation levels were calculated using the following formula:

$$2^{[-0.922 * (-C_{t_{HpaII-treated}} + C_{t_{HpaII-untreated}})]}$$

The coefficient of 0.922 was determined from the trend line shown in Extended Data Fig 3e.

The variability in C_t values is amplified when we apply the exponential transformation. For example, a variation of ± 0.5 in C_t values can lead to significant differences in the calculated fold change, particularly depending on the level of methylation.

For a sample with 100% methylation (where $-C_{t_{HpaII-treated}} + C_{t_{HpaII-untreated}} = 0$), the fold change can range from approximately 73% to 114%.

In contrast, for a sample with 1% methylation (where $-Ct_{\text{HpaII-treated}} + Ct_{\text{HpaII-untreated}} = 7$), the range is much narrower, from about 0.8% to 1.6%.

This illustrates that as the methylation level increases, the transformed fold change becomes more pronounced, resulting in larger apparent error bars.

- In Figure 1G, following the authors' description and also true in literature, the SNORD116 and IPW are part of long noncoding SNHG14 transcript. However, in the data presented, the expression level of SNORD116 is lower than IPW/SNHG14. That is unexpected since the SNORD116 has many copies and is processed RNAs from IPW or SNHG14. Also, a negative control with RT- for SNORD116 is necessary after the DNase treatment of RNA preparation since SNORD116 is intronless

(Response) We thank the reviewer for this valuable comment regarding the expression analysis of *SNORD116*, particularly the importance of ruling out genomic DNA contamination given that *SNORD116* is intronless.

As part of our standard RNA preparation protocol, we included DNase treatment to remove genomic DNA contaminants. We acknowledge, however, that this step was not explicitly described in the original Methods section. To clarify this point, we have added a corresponding description in Page 12, Line 360.

In our qPCR analysis of *SNORD116* and *IPW*, we presented the data as relative expression levels compared to healthy control iPSCs. In this case, we were not able to directly compare the relative expression values of *SNORD116* and *IPW*. Importantly, our results indicate that epigenome editing led to an upregulation of both genes to levels comparable to healthy control iPSCs.

As the reviewer suggested, we performed qPCR with RT(-) samples after DNase treatment and eliminated the possibility of genomic DNA contamination in cDNA.

RT-qPCR analysis of the expression of *SNORD116* in iPSCs (WT, n = 9 experiments, 3 iPSC lines; PWS, n = 6 experiments, 2 iPSC lines; gRNAs(-), n = 6 experiments, 6 clones from 2 iPSC lines; gRNAs(+), n = 6 experiments, 6 clones from 2 iPSC lines).

Additionally, we performed RNA-seq analysis of iPSCs and verified that *SNORD116* and *IPW* were upregulated by epigenome editing. These new analyses are included in Extended Data Fig 5, and we have added a corresponding description in Page 5, Line 131.

Extended Data Fig. 5 Transcriptomic analysis of epigenome-edited iPSCs
 a, Heatmap summarizing the expression of PWS-associated imprinted genes. b, RNA-seq read coverage mapped to the genomic region of *SNHG14*.

• 4) *MKRN3* promoter region in organoid didn't maintain demethylation (Fig 3E). However, *MKRN3* expression was unsilenced and increased expression from maternal chromosome. This result may suggest dCas9-Tet1 is not related to *MKRN3* reactivation and qPCR results may reflect DNA contamination. Or other mechanistic explanation that is important to understand the role of PWS-IC

(Response) We validated our qPCR results of *MKRN3* upregulation by checking RT(-) controls.

RT-qPCR analysis of the expression of *MKRN3* in organoids at d60 (TIG119, n = 3 experiments, 1 iPSC lines; WD39, n = 3 experiments, 1 iPSC lines; 2PWS8, n = 3 experiments, 1 iPSC lines; gRNAs(-), n = 3 experiments, 3 clones from 1 iPSC lines; gRNAs, n = 3 experiments, 3 clones from 1 iPSC lines).

Due to the relatively low coverage of our nanopore sequencing data mapped to the *MKRN3* region, we performed an additional methylation characterization of differentiated organoids using genomic qPCR after digestion with methylation-sensitive restriction enzymes. We found significant demethylation of the *MKRN3* promoter region in the epigenome-edited organoids, indicating that this promoter demethylation contributes to the upregulation of *MKRN3* expression. These findings are presented in Fig 3f, and we have added a corresponding description in Page 7, Line 205.

Fig. 3 Expression of neuron-specific imprinted genes after hypothalamic differentiation
 f, Methylation status of *MKRN3* region in organoids at d60, measured by genomic qPCR analysis following methylation-sensitive enzyme digestion (WD39, n = 3 differentiations, 1 iPSC line; gRNAs(-) dCas9-Suntag-TET1 (-), n = 3 differentiations, 1 iPSC line; gRNAs(-) dCas9-Suntag-TET1 (+), n = 3 differentiations, 1 clones from 1 iPSC lines; gRNAs(+) dCas9-Suntag-TET1 (+), n = 3 differentiations, 3 clones from 1 iPSC lines). One-way ANOVA with Tukey correction, F3, 8 = 22.79, ***P = 0.0009, **P = 0.0091, *P = 0.0495).

- As a positive control to compare the efficacy, authors may consider the DNMT inhibitor (ex;5-Aza) to see PEG expression from maternal chromosome in PWS iPSCs?

(Response) We treated PWS iPSCs with 5-Aza and observed an increase in *SNRPN* expression. However, since 5-Aza functions as a global DNA methylation inhibitor, its application in iPSCs was highly toxic, hindering proliferation and organoid differentiation. Additionally, it is important to highlight that 5-Aza treatment did not fully unsilence *SNRPN* expression to levels comparable to those of healthy control lines, whereas CRISPR-mediated epigenome editing achieved a comparable level of upregulation to that seen in healthy controls.

(Left) RT-qPCR analysis of the expression of *SNRPN* in iPSCs. 'd1' denotes the sample with 1 day treatment of 5-Aza.

(Right) Bright-field images of DMSO- or 5-Aza-treated iPSCs.

Reviewer #5

Nemoto et al has generated enhanced version of dCas9-Tet1 as a epigenome editing tool. They used Suntag to provide multiple platform to increase the possibility of gRNA-binding target sites for demethylation on CpGs. They applied the tool into iPSC derived from PWS patients with different types of deletions, UPD, and imprinting defect to unsilence the imprinted gene expression from maternal chromosome. They could show efficacy on large deletion of PWS mainly, rather than UPD. Overall, compared to the original ideas (Liu et al., 2016, Cell 167, 233–247; Nunez et al., 2021, Cell 184, 2503–2519), it seems like that its efficacy is enhanced but the edited single-cell derived clones and organoid definitely showed almost 100% rescued expression. I'd like to suggest that increased efficacy by initial transduction is meaningful for application to human, even if it is lenti-vial mediated delivery (lenti-virus is not appropriate to human, though). To prove the concept of principle as they suggest, the following questions should be addressed.

My major concerns are,

1) Considering the impact of dCas9-Tet1 with 5 guide RNA on wide range, (as my knowledge as well as authors mentioned, its covering target site for epigenome editing is 200 bp), demethylation on multiple loci, especially only the promoter region of each gene (SNRPN, MAGEL2, MKRN3) was shown in iPSCs. how could they recognize CpG sites on promoter only, not the whole CpG sites near the PWS-ICR? Could they provide methylation status profiling of MKRN3-SNORD115 region?

(Response) We thank the reviewer for raising this important question regarding the specificity of the demethylation effects observed following epigenome editing. We summarized the methylation difference between gRNAs(+) and gRNAs(-) iPSCs in Extended Data Fig 3i and Extended Data Fig 6f. Only the promoter regions of *SNRPN* and *MAGEL2* were clearly demethylated by epigenome editing. These findings suggest that the epigenome editing in our system regulated the methylation status of promoter regions of PWS-associated genes.

Extended Data Fig. 3 SNRPN expression and methylation change by epigenome editing
i, Methylation difference between gRNAs(+) and gRNAs(-) iPSCs in the genomic region of SNHG14.

Extended Data Fig. 6 Rescue of PWS-associated imprinted genes in epigenome-edited iPSCs
f, Methylation difference between gRNAs(+) and gRNAs(-) iPSCs in the genomic region of MKRN3–NDN.

2) Figure 1D, methylation status in PWS iPSC representing the maternal chromosome, is about 75%-120% , 150%. Could you explain the meaning of these variable value?

(Response) We thank the reviewer for pointing out the apparent methylation values exceeding 100% in Figure 1d. In methylation qPCR analysis, we utilized a strategy similar to $\Delta\Delta C_t$ method, which indeed resulted in larger error bars for samples with higher methylation levels. This increase in variability is primarily due to the logarithmic nature of the fold change calculations derived from the C_t values. As a result, the methylation levels may exceed 100% due to the mathematical properties of the transformation, not due to actual biological methylation levels exceeding 100%.

In our study, we employed the methylation-sensitive restriction enzyme HpaII to digest genomic DNA, allowing us to obtain C_t values for both HpaII-treated and untreated samples. The methylation levels were calculated using the following formula:

$$2^{[-0.922*(-C_{t_{\text{HpaII-treated}}} + C_{t_{\text{HpaII-untreated}}})]}$$

The coefficient of 0.922 was determined from the trend line shown in Extended Data Fig 3e.

The variability in C_t values is amplified when we apply the exponential transformation. For example, a variation of ± 0.5 in C_t values can lead to significant differences in the calculated fold change, particularly depending on the level of methylation.

For a sample with 100% methylation (where $-C_{t_{\text{HpaII-treated}}} + C_{t_{\text{HpaII-untreated}}} = 0$), the fold change can range from approximately 73% to 114%.

In contrast, for a sample with 1% methylation (where $-C_{t_{\text{HpaII-treated}}} + C_{t_{\text{HpaII-untreated}}} = 7$), the range is much narrower, from about 0.8% to 1.6%.

This illustrates that as the methylation level increases, the transformed fold change becomes more pronounced, resulting in larger apparent error bars.

3) in Figure 1G, following the authors' mentions, if the SNORD116 and IPW is part of SNRPN transcript (exactly, I would correct it as the part of SNHG14), the expression level of SNORD116 and IPW could be similar. In case of SNORD116, the variation of the each replicate is too broad range to argue the rescue the loss of gene expression.

(Response) We thank the reviewer for pointing out the inaccuracy in the original description of *SNORD116* and *IPW*. As correctly noted, *SNORD116* is processed from an intron of the *SNHG14* gene, and *IPW* is part of the long non-coding transcript of *SNHG14*, rather than originating from the intron of *SNRPN*. To address this, we updated the description of the genes in the Result section as follows:

(Page 5, line 132)

The expression of *SNORD116* and *IPW*, which are processed from the intron of the *SNHG14* gene and part of *SNHG14* transcript, respectively, was upregulated and comparable to the healthy control iPSCs (Fig. 1h, Extended Data Fig. 6a).

In our qPCR analysis of *SNORD116* and *IPW*, we presented the data as relative expression levels compared to healthy control iPSCs. In this case, we were not able to directly compare the relative expression values of *SNORD116* and *IPW*. Importantly, our results indicate that epigenome editing led to an upregulation of both genes to levels comparable to healthy control iPSCs.

To verify our qPCR results, we performed RNA-seq analysis of iPSCs and confirmed the upregulation of *SNORD116* and *IPW* expression by epigenome editing. These new analyses are included in Extended Data Fig 5, and we have added a corresponding description in Page 5, Line 131.

Extended Data Fig. 5 Transcriptomic analysis of epigenome-edited iPSCs
a, Heatmap summarizing the expression of PWS-associated imprinted genes. b, RNA-seq read coverage mapped to the genomic region of *SNHG14*.

4) *MKRN3* promoter region in organoid didn't maintain demethylation (Fig 3E). However, *MKRN3* expression was unsilenced and increased expression from maternal chromosome. This result may suggest dCas9-Tet1 is not related to *MKRN3* reactivation and qPCR results may reflect DNA contamination.

(Response) As the reviewer suggested, we validated our qPCR results of *MKRN3* upregulation by checking RT(-) controls.

RT-qPCR analysis of the expression of *MKRN3* in organoids at d60 (TIG119, n = 3 experiments, 1 iPSC lines; WD39, n = 3 experiments, 1 iPSC lines; 2PWS8, n = 3 experiments, 1 iPSC lines; gRNAs(-), n = 3 experiments, 3 clones from 1 iPSC lines; gRNAs, n = 3 experiments, 3 clones from 1 iPSC lines).

As our nanopore sequencing data has relatively low coverage mapped to the *MKRN3* region, we performed an additional methylation characterization of differentiated organoids using genomic qPCR after digestion with methylation-sensitive restriction enzymes. We found significant demethylation of the *MKRN3* promoter region in the epigenome-edited organoids, indicating that this promoter demethylation contributes to the upregulation of *MKRN3* expression. These findings are presented in Fig 3f, and we have added a corresponding description in Page 7, Line 205.

Fig. 3 Expression of neuron-specific imprinted genes after hypothalamic differentiation
 f, Methylation status of *MKRN3* region in organoids at d60, measured by genomic qPCR analysis following methylation-sensitive enzyme digestion (WD39, n = 3 differentiations, 1 iPSC line; gRNAs(-) dCas9-Suntag-TET1 (-), n = 3 differentiations, 1 iPSC line; gRNAs(-) dCas9-Suntag-TET1 (+), n = 3 differentiations, 1 clones from 1 iPSC lines; gRNAs(+/-) dCas9-Suntag-TET1 (+), n = 3 differentiations, 3 clones from 1 iPSC lines. One-way ANOVA with Tukey correction, F3, 8 = 22.79, ***P = 0.0009, **P = 0.0091, *P = 0.0495).

5) As a positive control, have the authors been tried to use DNMT inhibitor (ex;5-Aza) to see PEG expression from maternal chromosome in PWS iPSCs ?

(Response) We treated PWS iPSCs with 5-Aza and observed an increase in *SNRPN* expression. However, since 5-Aza functions as a global DNA methylation inhibitor, its application in iPSCs was highly toxic, hindering proliferation and organoid differentiation. Additionally, it is important to highlight that 5-Aza treatment did not fully unsilence *SNRPN* expression to levels comparable to those of healthy control lines, whereas CRISPR-mediated epigenome editing achieved a comparable level of upregulation to that seen in healthy controls.

(Left) RT-qPCR analysis of the expression of *SNRPN* in iPSCs. 'd1' denotes the sample with 1 day treatment of 5-Aza.

(Right) Bright-field images of DMSO- or 5-Aza-treated iPSCs.

6) NDN loci is closer to PWS-ICR than *MAGEL2*. However, *MAGEL2* was only unsilenced, not NDN. Authors suggest that *MAGEL2* expression is secondary demethylation governed by PWS-ICR. They may need solid mechanistic evidences to insist this model.

(Response) We thank the reviewer for this insightful comment regarding the selective reactivation of *MAGEL2*. To address the mechanism of *MAGEL2* reactivation, we analyzed the methylation dynamics during the epigenome editing process. We performed gRNA lentiviral infection and dCas9-Suntag-TET1 lipofection in PWS iPSCs and then maintained the cells in bulk without single-cell cloning, allowing us to observe the temporal pattern of methylation and gene expression change right after transfection (Extended Data Fig. 7a). We observed that PWS-ICR demethylation occurred immediately after dCas9-Suntag-TET1 transfection. In contrast, *MAGEL2* demethylation was observed subsequently, particularly at least a few passages after the transfection (Extended Data Fig. 7b), by which time dCas9-Suntag-TET1 was already eliminated (Extended Data Fig. 2b). We also detected a time lag of the expression upregulation between *SNRPN* and *MAGEL2* (Extended Data Fig. 7c).

These time lags of demethylation and gene expression suggest that the demethylation of *MAGEL2* is not a consequence of off-target effects from dCas9-Suntag-TET1. Instead, it implies that the demethylation of PWS-ICR subsequently regulates the methylation status of *MAGEL2*. These new analyses are included in Extended Data Fig 7a-c, and we have added a corresponding description in Page 6 Line 152.

Extended Data Fig. 7 Methylation dynamics of PWS-ICR/*SNRPN* and *MAGEL2* during epigenome editing
 a, Schedule of epigenome editing of iPSCs for methylation dynamics. b, Methylation status of PWS-ICR and *MAGEL2* region in epigenome-edited PWS iPSCs measured by genomic qPCR analysis following methylation-sensitive enzyme digestion (gRNAs(-), n = 3 experiments, 1 iPSC line; gRNAs(+), n = 3 experiments, 1 iPSC line). Lines denote fitted curves by LOESS regression. c, RT-qPCR analysis of the expression of *SNRPN* and *MAGEL2* (gRNAs(-), n = 3 experiments, 1 iPSC line; gRNAs(+), n = 3 experiments, 1 iPSC line). Lines denote fitted curves by LOESS regression.

Point-by-Point Responses to Reviewers

We thank the reviewers for their constructive feedback. We made modifications to the manuscript according to their suggestions. In these point-by-point responses, the original reviewer comments are shown in **black**, while our responses are shown in **blue**.

Reviewer #1: The authors have satisfactorily addressed my concerns. I support acceptance.

(Response) We thank the reviewer for the feedback and the positive evaluation of our work.

Reviewer #3: I co-reviewed this manuscript with one of the reviewers who provided the listed reports. This is part of the Nature Communications initiative to facilitate training in peer review and to provide appropriate recognition for Early Career Researchers who co-review manuscripts.

Reviewer #5: I co-reviewed this manuscript with one of the reviewers who provided the listed reports. This is part of the Nature Communications initiative to facilitate training in peer review and to provide appropriate recognition for Early Career Researchers who co-review manuscripts.

I appreciate the authors' efforts to address the reviewers' comments. I have one minor question: In the experiment of differentiated organoid, have you compared the *SNRPN* expression level between healthy control vs dCas9-Suntag-TET1 w/gRNA in 2PWS8?

(Response) This is a great suggestion. In fact, PWS iPSC-derived organoids exhibit ~1–5% of the *SNRPN* expression observed in healthy control organoids (Fig. 2f). Therefore, although epigenome editing resulted in a ~6-fold increase in *SNRPN* expression in differentiated organoids, this level remained below that of the healthy state. This limitation is, at least in part, due to the limited transfection efficiency of the dCas9-SunTag-TET1 and gRNAs. Accordingly, we have added the following statements in the Results section to clarify this limitation:

(Page 9, Line 261)

We observed a ~6-fold increase in *SNRPN* expression (Supplementary Fig. 12c), though it did not reach the levels observed in healthy control lines. These results suggest that epigenome editing in differentiating cells can partially reactivate *SNRPN*.

From the editors: We also invited R5 to have a look at their co-reviewer R4's comments as R4 was not able to review the revision. R5 believed that overall, you have addressed the reviewer's concern. However, they have one minor question regarding DNA methylation level in MAGEL2-SNRPN loci. Do you have any suggestion or insight related to the possibility of epigenetic drift in your organoids culture system following this paper (Franzen, J., Georgomanolis, T., Selich, A. et al. DNA methylation changes during long-term *in vitro* cell culture are caused by epigenetic drift. *Commun Biol* 4, 598 (2021)). Please describe briefly in the part of Discussion.

(Response) To address this, we have the following in the Discussion section of the manuscript:

(Page 10, Line 307)

Since epigenetic drift can occur during long-term *in vitro* culture and with aging *in vivo*²⁶, future studies are important to determine whether CRISPR-mediated epigenome editing provides stable imprinting control in the long term.

ROUND 2 REVIEWER 5 ATTACHMENT:

I appreciate the authors' efforts to address the reviewers' comments. I have one minor question: In the experiment of differentiated organoid, have you compared the SNRPN expression level between healthy control vs dCas9-Suntag-TET1 w/gRNA in 2PWS8?

Extended Data Fig. 12: Epigenome editing in organoids, related to Fig. 4